# Searching for Optimal Per-Coordinate Step-sizes with Multidimensional Backtracking

**Frederik Kunstner**   **Victor S. Portella**   **Mark Schmidt**[†]   **Nick Harvey**
{kunstner,victorsp,schmidtm,nickhar}@cs.ubc.ca
University of British Columbia       Canada CIFAR AI Chair (Amii)[†]

## Abstract

The backtracking line-search is an effective technique to automatically tune the step-size in smooth optimization. It guarantees similar performance to using the theoretically optimal step-size. Many approaches have been developed to instead tune per-coordinate step-sizes, also known as diagonal preconditioners, but none of the existing methods are provably competitive with the optimal per-coordinate step-sizes. We propose *multidimensional backtracking*, an extension of the backtracking line-search to find good diagonal preconditioners for smooth convex problems. Our key insight is that the gradient with respect to the step-sizes, also known as hyper-gradients, yields separating hyperplanes that let us search for good preconditioners using cutting-plane methods. As black-box cutting-plane approaches like the ellipsoid method are computationally prohibitive, we develop an efficient algorithm tailored to our setting. Multidimensional backtracking is provably competitive with the best diagonal preconditioner and requires no manual tuning.

## 1 Introduction

When training machine learning models, tuning the hyperparameters of the optimizer is often a major challenge. For example, finding a reasonable step-size hyperparameter for gradient descent typically involves trial-and-error or a costly grid search. In smooth optimization, a common approach to set the step-size without user input is a backtracking line-search: start with a large step-size, and decrease it when it is too big to make sufficient progress. For ill-conditioned problems, however, there are limits to the improvement achievable by tuning the step-size. Per-coordinate step-sizes—also known as diagonal preconditioners—can drastically improve performance. Many approaches have been developed to automatically tune per-coordinate step-sizes. Those are often described as "adaptive" methods, but the meaning of this term varies widely, from describing heuristics that set per-coordinate step-sizes, to ensuring performance guarantees as if a particular property of the problem were known in advance. Yet, even on the simplest case of a smooth and strongly convex deterministic problem where a good fixed diagonal preconditioner exists (i.e., one that reduces the condition number), none of the existing adaptive methods are guaranteed to find per-coordinate step-sizes that improve the convergence rate. We discuss approaches to adaptive methods in the next section.

**Contribution.** We propose *multidimensional backtracking*, an extension of the standard backtracking line-search to higher dimension, to automatically find good per-coordinate step-sizes. Our method recovers the convergence rate of gradient descent with the *optimal preconditioner* for the problem, up to a $\sqrt{2d}$ factor where $d$ is the number of coordinates. This is a direct generalization of the line-search guarantee, with a penalty depending on dimension due to the extra degrees of freedom, as expected.

### 1.1 Adaptive step-sizes and preconditioning methods

**Adaptive and parameter-free methods in online learning** are an example where *adaptive methods* have a well-defined meaning. AdaGrad (McMahan and Streeter, 2010; Duchi et al., 2011) and Coin Betting (Orabona and Pál, 2016; Orabona and Tommasi, 2017) can adapt to problem-specific constants without user input and have strong guarantees, even in the *adversarial* setting. However,

37th Conference on Neural Information Processing Systems (NeurIPS 2023).

this resilience to adversaries is a double-edged sword; to satisfy this definition of adaptivity, AdaGrad uses monotonically decreasing step-sizes. While AdaGrad still converges at the desired asymptotic rate on smooth, Lipschitz functions (Ward et al., 2019; Li and Orabona, 2019), its performance can be worse than plain gradient descent. This motivated investigations of workarounds to avoid the monotonically decreasing updates, including augmenting the update with an increasing step-size schedule (Agarwal et al., 2020), a line-search (Vaswani et al., 2020), or modifying the update to the preconditioner (Defazio et al., 2022). Methods commonly used in deep learning, such as RMSProp and Adam (Hinton et al., 2012; Kingma and Ba, 2015), are often motivated as *adaptive* by analogy to AdaGrad, but without decreasing step-sizes (e.g., Défossez et al., 2022, §4.3). This change is crucial for their practical performance, but nullifies their online-learning adaptivity guarantees.

**Adaptive gain and hypergradient heuristics.** Many heuristics that tune the hyperparameters of the optimization procedure use the gradient with respect to the hyperparameters, or *hypergradients* (Maclaurin et al., 2015). Methods have been proposed to tune the step-size (Masse and Ollivier, 2015), a preconditioner (Moskovitz et al., 2019), any hyperparameter (Baydin et al., 2018), or to maintain a model of the objective (Bae et al., 2022). "Stacking" such optimizers recursively has been shown to reduce the dependency on user-specified hyperparameters in practice (Chandra et al., 2022). This idea pre-dates the hypergradient nomenclature; Kesten (1958) presents a method to update the step-size based on the sign of successive gradients, and Saridis (1970) presents a control perspective for per-coordinate step-sizes, which can be cast as a hypergradient update to a diagonal preconditioner.[1] This approach has led to *adaptive gain* methods such as Delta-Delta and variants (Barto and Sutton, 1981; Jacobs, 1988; Silva and Almeida, 1990; Sutton, 1992a,b), and further developed using the sign of the hypergradient (Riedmiller and Braun, 1993), full-matrix updates (Almeida et al., 1999), a larger history (Plagianakos et al., 2001), updates in log-space (Schraudolph, 1999; Schraudolph et al., 2005), heuristics to adjust the outer step-size (Mahmood et al., 2012), or multiplicative weight updates (Amid et al., 2022). While showing promising practical performance in some settings, existing methods are often motivated from intuition rather than a formal definition of adaptivity, giving no guarantee that the tuned method will converge faster, if at all. Indeed, hypergradient methods are often unstable, and may require as much manual tuning as the original optimizer they are intended to tune.

**Second-order methods.** A classical approach to preconditioning is to use second-order information, as in Newton's method or its regularized variants (e.g., Nesterov and Polyak, 2006). To avoid the load of computing and inverting the Hessian, quasi-Newton methods (Dennis and Moré, 1977) such as L-BFGS (Liu and Nocedal, 1989) fit an approximate Hessian using the secant equation. Variants using diagonal approximations have also been proposed, framed as Quasi-Cauchy, diagonal BFGS, or diagonal Barzilai-Borwein methods (Zhu et al., 1999; Andrei, 2019; Park et al., 2020), while other methods use the diagonal of the Hessian (LeCun et al., 2012; Yao et al., 2021). Some second-order and quasi-Newton methods converge super linearly (although not the diagonal or limited memory variants used in practice), but those guarantees only hold locally when close to the minimum. To work when far from a solution, those methods require "globalization" modifications, such as regularization or a line-search. Unfortunately, analyses of second-order methods do not capture the global benefit of preconditioning and instead lead to worse rates than gradient descent, as in the results of Byrd et al. (2016, Cor. 3.3), Bollapragada et al. (2018, Thm. 3.1), Meng et al. (2020, Thm. 1), Yao et al. (2021, Apx.), Berahas et al. (2022, Thm. 5.2), or Jahani et al. (2022, Thm. 4.9).

**Line-searches.** Adaptivity in smooth optimization is most closely related to line-searches. The standard guarantee for gradient descent on an $L$-smooth function requires a step-size of $1/L$, but $L$ is typically unknown. The backtracking line-search based on the Armijo condition (Armijo, 1966) approximately recovers this convergence guarantee by starting with a large step-size, and backtracking; halving the step-size whenever it does not yield sufficient improvement. However, line-searches are often overlooked in the discussion of adaptive methods, as they do not provide a way to set more than a scalar step-size. While line-searches can be shown to work in the stochastic overparameterized setting and have been applied to train neural networks (Vaswani et al., 2019), improvements beyond backtracking have been limited. Additional conditions (Wolfe, 1969), non-monotone relaxations (Grippo et al., 1986), or solving the line-search to higher precision (Moré and Thuente, 1994) can improve the performance in practice, but even an exact line-search cannot improve the convergence rate beyond what is achievable with a fixed step-size (Klerk et al., 2017).

---

[1]The hypergradient with respect to a diagonal preconditioner $\mathbf{P} = \mathrm{Diag}(\mathbf{p})$ is, by the chain rule, the element-wise product ($\odot$) of subsequent gradients, $-\nabla_{\mathbf{P}} f(\mathbf{x} - \mathrm{Diag}(\mathbf{p})\nabla f(\mathbf{x})) = \nabla f(\mathbf{x}) \odot \nabla f(\mathbf{x} - \mathrm{Diag}(\mathbf{p})\nabla f(\mathbf{x}))$.

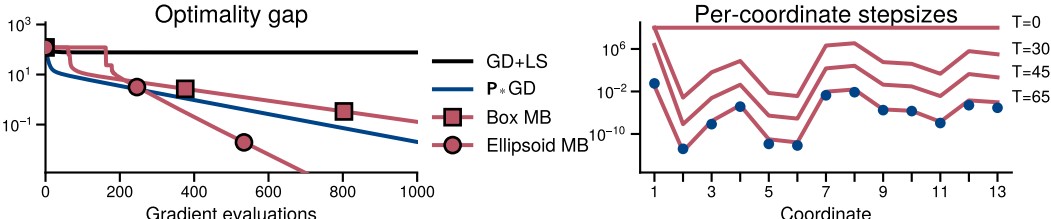

Figure 1: **Multidimensional backtracking can find the optimal diagonal preconditioner.** Example on a linear regression where the optimal preconditioner can be computed. **Left:** Performance of Gradient Descent (GD), optimally preconditioned GD ($\mathbf{P}_*$GD) with a line-search (+LS), and Multidimensional Backtracking (MB) with the strategies in Section 5. The ellipsoid variant can outperform the *globally* optimal preconditioner by selecting preconditioners that leads to more *local* progress. **Right:** Optimal per-coordinate step-sizes (•) and the ones found by MB (box) across iterations.

## 1.2 Summary of main result: adaptivity to the optimal preconditioner

Our approach is inspired by the work discussed above, but addresses the following key limitation: none of the existing methods attain better global convergence rates than a backtracking line-search. Moreover, this holds even on smooth convex problems for which a good preconditioner exists.

We generalize the backtracking line-search to handle per-coordinate step-sizes and find a good preconditioner. As in quasi-Newton methods, we build a preconditioner based on first-order information. However, instead of trying to approximate the Hessian using past gradients, our method searches for a preconditioner that minimizes the objective function at the next step. Our convergence result depends on the best rate achievable by an *optimal diagonal preconditioner*, similarly to how methods in online learning are competitive against the best preconditioner in hindsight. However, our notion of optimality is tailored to smooth strongly-convex problems and does not require decreasing step-sizes as in AdaGrad. Our update to the preconditioner can be interpreted as a hypergradient method, but instead of a heuristic update, we develop a cutting-plane method that uses hypergradients to guarantee a good diagonal preconditioner. Our main theoretical contribution is summarized below.

**Theorem 1.1** (Informal). *On a smooth, strongly-convex function $f$ in $d$ dimensions, steps accepted by multidimensional backtracking guarantee the following progress*

$$f(\mathbf{x}_{t+1}) - f(\mathbf{x}_*) \le \left(1 - \frac{1}{\sqrt{2d}}\frac{1}{\kappa_*}\right)\left(f(\mathbf{x}_t) - f(\mathbf{x}_*)\right),$$

*where $\kappa_*$ is the condition number achieved by the optimal preconditioner defined in Section 2. The number of backtracking steps is at most linear in $d$ and logarithmic in problem-specific constants.*

Multidimensional backtracking finds per-coordinate step-sizes that lead to a provable improvement over gradient descent on badly conditioned problems that can be improved by diagonal preconditioning, i.e., if the condition number of $f$ is at least $\sqrt{2d} \cdot \kappa_*$. Moreover, this guarantee is worst-case, and multidimensional backtracking can outperform the globally optimal preconditioner by finding a better *local* preconditioner, as illustrated on an ill-conditioned linear regression problem in Figure 1.

To find a competitive diagonal preconditioner, we view backtracking line-search as a cutting-plane method and generalize it to higher dimensions in Section 3. In Section 4 we show how to use hypergradients to find separating hyperplanes in the space of preconditioners, and in Section 5 we develop an efficient cutting-plane methods tailored to the problem. In Section 6, we illustrate the method through preliminary experiments and show it has consistent performance across problems.

**Notation.** We use standard font weight $d$, $n$, $\alpha$ for scalars, bold $\mathbf{x}$, $\mathbf{y}$ for vectors, and capital bold $\mathbf{P}$, $\mathbf{A}$ for matrices. We use $\mathbf{p}[i]$ for the $i$-th entry of $\mathbf{p}$, use $\odot$ for element-wise multiplication, and define $\mathbf{p}^2 := \mathbf{p} \odot \mathbf{p}$. We use $\mathbf{P} = \mathrm{Diag}(\mathbf{p})$ to denote the diagonal matrix with diagonal $\mathbf{p}$, and $\mathbf{p} = \mathrm{diag}(\mathbf{P})$ to denote the vector of diagonal entries of $\mathbf{P}$. We say $\mathbf{A}$ is larger than $\mathbf{B}$, $\mathbf{A} \succeq \mathbf{B}$, if $\mathbf{A} - \mathbf{B}$ is positive semidefinite. If $\mathbf{A} = \mathrm{Diag}(\mathbf{a})$, $\mathbf{B} = \mathrm{Diag}(\mathbf{b})$, the ordering $\mathbf{A} \succeq \mathbf{B}$ is equivalent to $\mathbf{a}[i] \ge \mathbf{b}[i]$ for all $i$, which we write $\mathbf{a} \ge \mathbf{b}$. We use $\mathbf{I}$ for the identity matrix and $\mathbf{1}$ for the all-ones vector.

## 2 Optimal preconditioning and sufficient progress

Consider a twice-differentiable function $f \colon \mathbb{R}^d \to \mathbb{R}$ that is $L$-smooth and $\mu$-strongly convex,[2] i.e.,

$$\mu \tfrac{1}{2} \|\mathbf{x} - \mathbf{y}\|^2 \leq f(\mathbf{y}) - f(\mathbf{x}) - \langle \nabla f(\mathbf{x}), \mathbf{y} - \mathbf{x} \rangle \leq L \tfrac{1}{2} \|\mathbf{y} - \mathbf{x}\|^2, \quad \text{for all } \mathbf{x}, \mathbf{y},$$

or $\mu \mathbf{I} \preceq \nabla^2 f(\mathbf{x}) \preceq L\mathbf{I}$ for all $\mathbf{x}$. We measure the quality of a preconditioner $\mathbf{P}$ by how well $\mathbf{P}^{-1}$ approximates the Hessian $\nabla^2 f(\mathbf{x})$. We define an *optimal diagonal preconditioner* for $f$ as

$$\mathbf{P}_* \in \arg\min_{\mathbf{P} \succeq 0, \text{diagonal}} \kappa \qquad \text{such that} \qquad \tfrac{1}{\kappa} \mathbf{P}^{-1} \preceq \nabla^2 f(\mathbf{x}) \preceq \mathbf{P}^{-1} \text{ for all } \mathbf{x}, \qquad (1)$$

and denote by $\kappa_*$ the optimal $\kappa$ above. A related and known measure for the convergence rate of preconditioned methods is $\kappa(\mathbf{P}^{1/2} \nabla^2 f(\mathbf{x}) \mathbf{P}^{1/2})$ (Bertsekas, 1999, §1.3.2). Moreover, (1) reduces to the definition of optimal preconditioning for linear systems (Jambulapati et al., 2020; Qu et al., 2022) when $f$ is quadratic. Alternatively, the optimal preconditioner can be viewed as the matrix $\mathbf{P}_*$ such that $f$ is 1-smooth and maximally strongly-convex in the norm $\|\mathbf{x}\|_{\mathbf{P}^{-1}}^2 = \langle \mathbf{x}, \mathbf{P}_*^{-1} \mathbf{x} \rangle$,

$$\tfrac{1}{\kappa_*} \tfrac{1}{2} \|\mathbf{x} - \mathbf{y}\|_{\mathbf{P}_*^{-1}}^2 \leq f(\mathbf{y}) - f(\mathbf{x}) - \langle \nabla f(\mathbf{x}), \mathbf{y} - \mathbf{x} \rangle \leq \tfrac{1}{2} \|\mathbf{y} - \mathbf{x}\|_{\mathbf{P}_*^{-1}}^2, \quad \text{for all } \mathbf{x}, \mathbf{y}. \qquad (2)$$

Similar definitions of smoothness and strong-convexity relative to a matrix are common in coordinate descent methods (e.g., Qu et al., 2016; Safaryan et al., 2021), where the matrices are assumed to be known a priori. If we knew $\mathbf{P}_*$, preconditioned gradient descent using $\mathbf{P}_*$ would converge at the rate

$$f(x - \mathbf{P}_* \nabla f(\mathbf{x})) - f(\mathbf{x}_*) \leq \left(1 - \tfrac{1}{\kappa_*}\right)(f(\mathbf{x}) - f(\mathbf{x}_*)),$$

where $\mathbf{x}_*$ minimizes $f$. We do not know $\mathbf{P}_*$ and will be searching for a good approximation.

For the standard backtracking line-search on $L$-smooth functions, the goal is to find a step-size that works as well as $1/L$ without knowledge of $L$. To do so, we can start with a large step-size $\alpha \gg 1/L$ and check the *Armijo condition*: the step-size $\alpha$ makes progress as if $f$ were $1/\alpha$-smooth, that is,

$$f(\mathbf{x} - \alpha \nabla f(\mathbf{x})) \leq f(\mathbf{x}) - \alpha \tfrac{1}{2} \|\nabla f(\mathbf{x})\|^2. \qquad (3)$$

If the condition is satisfied, we take the step $\mathbf{x} - \alpha \nabla f(\mathbf{x})$. By the descent lemma, (Bertsekas, 1999, A.24), the condition is satisfied if $\alpha \leq 1/L$. So if the condition fails, we know $\alpha$ is too large and can decrease $\alpha$. For diagonal preconditioners, the Armijo condition checks whether the preconditioner makes sufficient progress in the norm induced by $\mathbf{P}^{-1}$, as if $f$ were 1-smooth in Equation (2), that is,

$$f(\mathbf{x} - \mathbf{P} \nabla f(\mathbf{x})) \leq f(\mathbf{x}) - \tfrac{1}{2} \|\nabla f(\mathbf{x})\|_{\mathbf{P}}^2. \qquad (4)$$

As with a scalar step-size, sufficient progress holds for any matrix $\mathbf{P}$ that satisfies $\nabla^2 f(\mathbf{x}) \preceq \mathbf{P}^{-1}$.

## 3 Multidimensional Backtracking

The typical presentation of the backtracking line-search maintains a step-size and decreases it when the Armijo condition fails (e.g., Nocedal and Wright, 1999, Alg 3.1). We instead take the following non-standard view, which generalizes more naturally to high dimension; as maintaining a set *containing* the optimal step-size, and using bisection to narrow down the size of the set. Starting with an interval $\mathcal{S} = [0, \alpha_{\max}]$ containing $1/L$, we pick a candidate step-size $\alpha$ by "backtracking" by $\gamma < 1$ from the largest step-size in $\mathcal{S}$, taking $\alpha = \gamma \alpha_{\max}$ to balance two properties;

1. **Large progress:** If the candidate step-size satisfies the Armijo condition and the step is accepted, the value of $f$ decreases proportionally to $\alpha$ as in (3). To maximize the progress, $\gamma$ should be large.
2. **Volume shrinkage:** If the candidate step-size fails the Armijo condition, we learn that $\alpha > 1/L$ and can cut the interval to $\mathcal{S}' = [0, \gamma \alpha_{\max}]$. To ensure the interval shrinks fast, $\gamma$ should be small.

Taking $\gamma = 1/2$ balances both properties; $\alpha$ is at least $1/2$ as large as any step-size in $\mathcal{S}$, and we can halve the interval if the Armijo condition fails. We do not use $\alpha_{\max}$ as a candidate since, although the largest in $\mathcal{S}$, it would give no information to update the interval in case it failed the Armijo condition.

For multidimensional backtracking, we can check whether a candidate preconditioner yields sufficient progress with Equation (4) instead of the Armijo condition, and replace the intervals by sets of diagonal preconditioners. The high-level pseudocode is given in Figure 2, where each iteration either leads to an improvement in function value or shrinks the sets of potential step-sizes/preconditioners.

---

[2]While we use strong-convexity and twice-differentiability of $f$ to define the optimal preconditioner, those assumptions can be relaxed to only rely on the PL inequality (Polyak, 1963; Łojasiewicz, 1963) (see Appendix B).

| **Backtracking line-search** | **Multidimensional Backtracking** |
|---|---|

**Backtracking line-search**

**Input:** starting point $\mathbf{x}_0$, backtracking coefficient $\gamma$, set $\mathcal{S}_0 = [0, \alpha_0^{\max}]$ containing the optimal step-size $1/L$.

Iterate for $t$ in $0, 1, ..., T$
    Pick step-size $\alpha_t = \gamma \alpha_t^{\max}$
    If $(\mathbf{x}_t, \alpha_t)$ satisfy the Armijo condition (3)
        Accept $\mathbf{x}_{t+1} = \mathbf{x}_t - \alpha_t \nabla f(\mathbf{x}_t)$
        Keep max step-size $\alpha_{t+1}^{\max} = \alpha_t^{\max}$
    Otherwise,
        Don't move, $\mathbf{x}_{t+1} = \mathbf{x}_t$
        Cut max step-size $\alpha_{t+1}^{\max} = \gamma \alpha_t^{\max}$
**Output:** $\mathbf{x}_T$

**Multidimensional Backtracking**

**Input:** starting point $\mathbf{x}_0$, backtracking coefficient $\gamma$, set $\mathcal{S}_0$ of preconditioners containing the optimal $\mathbf{P}_*$.

Iterate for $t$ in $0, 1, ..., T$
    Pick step-sizes $\mathbf{P}_t = \text{CANDIDATE}(\mathcal{S}_t, \gamma, \mathbf{x}_t)$    (†)
    If $(\mathbf{x}_t, \mathbf{P}_t)$ satisfy the Armijo condition (4)
        Accept $\mathbf{x}_{t+1} = \mathbf{x}_t - \mathbf{P}_t \nabla f(\mathbf{x}_t)$
        Keep set $\mathcal{S}_{t+1} = \mathcal{S}_t$
    Otherwise,
        Don't move, $\mathbf{x}_{t+1} = \mathbf{x}_t$
        Cut set $\mathcal{S}_{t+1} = \text{CUT}(\mathcal{S}_t, \mathbf{x}_t, \mathbf{P}_t)$    (†)
**Output:** $\mathbf{x}_T$

Figure 2: **Pseudocode for the backtracking line-search and multidimensional backtracking.** We view backtracking as maintaining a set of step-sizes, testing one at each iteration that either make progress on $f$ or reduce the size of the set. Steps marked by (†), are the subject of Sections 3–5.

To complete the algorithm, we need to define the steps marked as (†) to select preconditioners that lead to large progress when the step is accepted, while significantly reducing the search space when the preconditioner does not yield sufficient progress. For computational efficiency, we want methods that take $O(d)$ time and memory like plain gradient descent.

### 3.1 Guaranteed progress competitive with the optimal preconditioner

We start by formalizing the progress guarantee. If $\mathbf{P}_t$ satisfies the Armijo condition (4) at $\mathbf{x}_t$, the function value decreases by at least $\|\nabla f(\mathbf{x}_t)\|_{\mathbf{P}_t}^2$. If we can guarantee that $\|\nabla f(\mathbf{x}_t)\|_{\mathbf{P}_t}^2 \geq \gamma \|\nabla f(\mathbf{x}_t)\|_{\mathbf{P}_*}^2$ for some $\gamma > 0$, we can recover the convergence rate of gradient descent preconditioned with $\mathbf{P}_*$ up to a factor of $\gamma$. However, we do not know $\mathbf{P}_*$, but know a set $\mathcal{S}_t$ that contains preconditioners we have not yet ruled out, including $\mathbf{P}_*$. To guarantee that $\mathbf{P}_t$ is competitive with $\mathbf{P}_*$, we can enforce that $\mathbf{P}_t$ is competitive with *all* the preconditioners in $\mathcal{S}_t$, as captured by the following definition.

**Definition 3.1** ($\gamma$-competitive candidate preconditioners)**.** *A matrix* $\mathbf{P}_t \in \mathcal{S}_t$ *is* $\gamma$-competitive in $\mathcal{S}_t$, *for a gradient* $\nabla f(\mathbf{x}_t)$, *if* $\|\nabla f(\mathbf{x}_t)\|_{\mathbf{P}_t}^2 \geq \gamma \|\nabla f(\mathbf{x}_t)\|_{\mathbf{Q}}^2$ *for any* $\mathbf{Q} \in \mathcal{S}_t$.

If $\mathbf{P}_t$ is $\gamma$-competitive, then it is competitive with $\mathbf{P}_*$ as $\max_{\mathbf{Q} \in \mathcal{S}_t} \|\nabla f(\mathbf{x}_t)\|_{\mathbf{Q}}^2 \geq \|\nabla f(\mathbf{x}_t)\|_{\mathbf{P}_*}^2$. However, this is a strong requirement. To illustrate what competitive ratios are attainable, we show in Appendix B that even the optimal preconditioner $\mathbf{P}_*$ might only be $1/d$-competitive, as other preconditioners can lead to more *local* progress depending on $\nabla f(\mathbf{x}_t)$, whereas $\mathbf{P}_*$ is a fixed *global* optimal preconditioner. This also suggests that selecting a preconditioner that guarantees more local progress may lead to better performance, which we take advantage of to ensure a $\gamma = 1/\sqrt{2d}$ competitive ratio.

To see how to ensure a competitive ratio, consider the case where $\mathcal{S}$ contains diagonal preconditioners whose diagonals come from the box $\mathcal{B}(\mathbf{b}) := \{ \mathbf{p} \in \mathbb{R}_{\geq 0}^d : \mathbf{p} \leq \mathbf{b} \}$. To select a candidate preconditioner that is $\gamma$-competitive in $\mathcal{S}$, we can backtrack from the largest vector in $\mathcal{B}(\mathbf{b})$ by some constant $\gamma < 1$, and take $\mathbf{P} = \gamma \, \text{Diag}(\mathbf{b})$. While a large $\gamma$ leads to more progress when the step is accepted, we will see that we need a small $\gamma$ to ensure the volume shrinks when the step is rejected.

We can obtain the convergence rate of Theorem 1.1 depending on $\gamma$ and the optimal preconditioned condition number $\kappa_*$ if we ensure $\mathbf{P}_* \in \mathcal{S}_t$ and that $\mathbf{P}_t$ is $\gamma$-competitive for all $t$.

**Proposition 3.2.** *Let* $\mathbf{P}_*, \kappa_*$ *be an optimal preconditioner and condition number for* $f$ (1). *If the set* $\mathcal{S}_t$ *from the algorithm in Figure 2 contains* $\mathbf{P}_*$, *and* $\mathbf{P}_t \in \mathcal{S}_t$ *is* $\gamma$-competitive (Definition 3.1), *then*

$$f(\mathbf{x}_{t+1}) - f(\mathbf{x}_*) \leq \left(1 - \frac{\gamma}{\kappa_*}\right)(f(\mathbf{x}_t) - f(\mathbf{x}_*))$$

*whenever the candidate step leads to sufficient progress and is accepted.*

*Proof.* The proof relies on three inequalities. (a) The iterate $\mathbf{x}_{t+1}$ yields sufficient progress (Eq. 4), (b) any accepted preconditioner $\mathbf{P}_t$ is $\gamma$-competitive in $\mathcal{S}_t$ and thus with $\mathbf{P}_*$, and (c) $f$ is $1/\kappa_*$-strongly convex in $\|\cdot\|_{\mathbf{P}^{-1}}$, which implies $\kappa_* \frac{1}{2} \|\nabla f(\mathbf{x}_t)\|_{\mathbf{P}_*}^2 \geq f(\mathbf{x}_t) - f(\mathbf{x}_*)$. Combining those yields

$$f(\mathbf{x}_{t+1}) \overset{(a)}{\leq} f(\mathbf{x}_t) - \frac{1}{2}\|\nabla f(\mathbf{x}_t)\|_{\mathbf{P}_t}^2 \overset{(b)}{\leq} f(\mathbf{x}_t) - \gamma \frac{1}{2}\|\nabla f(\mathbf{x}_t)\|_{\mathbf{P}_*}^2 \overset{(c)}{\leq} f(\mathbf{x}_t) - \frac{\gamma}{\kappa_*}(f(\mathbf{x}_t) - f(\mathbf{x}_*)).$$

Subtracting $f(\mathbf{x}_*)$ on both sides yields the contraction guarantee. $\qquad\square$

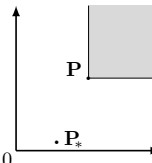 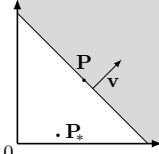 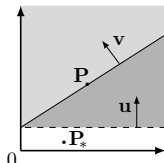

(a) Failing the Armijo condition cuts the interval in half in one dimension, but only removes $1/2^d$ of the volume in $d$ dimensions.

(b) Half-space $\mathcal{H}_>(\mathbf{v})$ obtained by using the hypergradient when failing the Armijo condition at $\mathbf{P}$ in Proposition 4.2.

(c) Stronger half-space $\mathcal{H}_>(\mathbf{u})$ described by Proposition 4.3, removing $\mathbf{P}' \succ \mathbf{P}$ for any $\mathbf{P}$ ruled out by $\mathcal{H}_>(\mathbf{v})$ in Proposition 4.2.

Figure 3: **Lack of information from the ordering and separating hyperplanes.**

## 4 Separating hyperplanes in higher dimensions

In one dimension, if the step-size $\alpha$ does not satisfy the sufficient progress condition (3), we know $\alpha > 1/L$ and can rule out any $\alpha' \geq \alpha$. We are looking for a generalization to higher dimensions: if the queried preconditioner fails the sufficient progress condition, we should be able to discard all larger preconditioners. The notion of *valid* preconditioners formalizes this idea.

**Definition 4.1** (Valid preconditioner). *A preconditioner $\mathbf{P}$ is* valid *if $\mathbf{P}^{1/2}\nabla^2 f(\mathbf{x})\mathbf{P}^{1/2} \preceq \mathbf{I}$ for all $\mathbf{x}$, which guarantees that $\mathbf{P}$ satisfies the sufficient progress (4) condition, and* invalid *otherwise.*

Validity is a global property: a preconditioner $\mathbf{P}$ might lead to sufficient progress locally but still be invalid. Using the partial order, if $\mathbf{P}$ is invalid then any preconditioner $\mathbf{P}' \succeq \mathbf{P}$ is also invalid. However, this property alone only discards an exceedingly small portion of the feasible region in high dimensions. Consider the example illustrated in Figure 3a: if the diagonals are in a box $\mathcal{B}(\mathbf{b})$, the fraction of volume discarded in this way if $(1/2)\operatorname{Diag}(\mathbf{b})$ is invalid is only $1/2^d$.

To efficiently search for valid preconditioners, we show that if $f$ is convex, then the gradient of the sufficient progress condition gives a *separating hyperplane* for valid preconditioners. That is, it gives a vector $\mathbf{u} \in \mathbb{R}^d$ such that if $\mathbf{p} \in \mathbb{R}^d_{\geq 0}$ satisfies $\langle \mathbf{u}, \mathbf{p} \rangle > 1$, then $\operatorname{Diag}(\mathbf{p})$ is invalid, as illustrated in Figure 3b. We use the following notation to denote normalized half-spaces:

$$\mathcal{H}_>(\mathbf{u}) := \{\, \mathbf{p} \in \mathbb{R}^d_{\geq 0} : \langle \mathbf{u}, \mathbf{p} \rangle > 1\} \qquad \text{and} \qquad \mathcal{H}_\leq(\mathbf{u}) := \{\mathbf{p} \in \mathbb{R}^d_{\geq 0} : \langle \mathbf{u}, \mathbf{p} \rangle \leq 1\}.$$

**Proposition 4.2** (Separating hyperplane in preconditioner space). *Suppose $\mathbf{Q} = \operatorname{Diag}(\mathbf{q}) \succ 0$ does not lead to sufficient progress (4) at $\mathbf{x}$, and let $h(\mathbf{q})$ be the gap in the sufficient progress condition,*

$$h(\mathbf{q}) := f(\mathbf{x} - \mathbf{Q}\nabla f(\mathbf{x})) - f(\mathbf{x}) + \tfrac{1}{2}\|\nabla f(\mathbf{x})\|^2_\mathbf{Q} > 0.$$

*Then $\operatorname{Diag}(\mathbf{p})$ for any $\mathbf{p}$ in the following half-space satisfies $h(\mathbf{p}) > 0$ and is also invalid,*

$$\{\, \mathbf{p} \in \mathbb{R}^d : \langle \nabla h(\mathbf{q}), \mathbf{p} \rangle > \langle \nabla h(\mathbf{q}), \mathbf{q} \rangle - h(\mathbf{q})\}, \tag{5}$$

*This half-space is equal to $\mathcal{H}_>(\mathbf{v})$ with $\mathbf{v}$ given by $\mathbf{v} = \nabla h(\mathbf{q})/(\langle \nabla h(\mathbf{q}), \mathbf{q} \rangle - h(\mathbf{q}))$, or*

$$\mathbf{v} := \frac{(\tfrac{1}{2}\mathbf{g} - \mathbf{g}^+) \odot \mathbf{g}}{f(\mathbf{x}) - \langle \mathbf{g}^+, \mathbf{Q}\mathbf{g} \rangle - f(\mathbf{x}^+)}, \qquad \text{with} \qquad \begin{cases} \mathbf{x}^+ := \mathbf{x} - \mathbf{Q}\nabla f(\mathbf{x}), \\ (\mathbf{g}, \mathbf{g}^+) := (\nabla f(\mathbf{x}), \nabla f(\mathbf{x}^+)). \end{cases} \tag{6}$$

*Proof idea.* If $f$ is convex, then $h$ also is. Convexity guarantees that $h(\mathbf{p}) \geq h(\mathbf{q}) + \langle \nabla h(\mathbf{q}), \mathbf{p} - \mathbf{q} \rangle$ for any $\mathbf{p}$. A sufficient condition for $h(\mathbf{p}) > 0$, which means $\mathbf{p}$ is invalid, is whether $h(\mathbf{q}) + \langle \nabla h(\mathbf{q}), \mathbf{p} - \mathbf{q} \rangle > 0$ holds. Reorganizing yields Equation (5), and Equation (6) expresses the half-space in normalized form, $\mathcal{H}_>(\mathbf{v})$, expanding $h$ in terms of $f$, its gradients, and $\mathbf{Q}$. □

The half-space in Proposition 4.2 is however insufficient to find good enough cutting-planes, as it uses convexity to invalidate preconditioners but ignores the ordering that if $\mathbf{P}$ is invalid, any $\mathbf{P}' \succeq \mathbf{P}$ is also invalid. If such preconditioners are not already ruled out by convexity, we can find a stronger half-space by removing them, as illustrated in Figure 3c. We defer proofs to Appendix C.

**Proposition 4.3** (Stronger hyperplanes). *If $\mathcal{H}_>(\mathbf{v})$ is a half-space given by Proposition 4.2, then $\mathcal{H}_>(\mathbf{u})$ where $\mathbf{u} := \max\{\mathbf{v}, 0\}$ element-wise is a stronger half-space in the sense that $\mathcal{H}_>(\mathbf{v}) \subseteq \mathcal{H}_>(\mathbf{u})$, and $\mathcal{H}_>(\mathbf{u})$ contains only invalid preconditioners.*

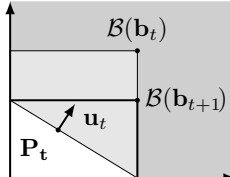
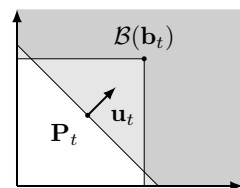

(a) **Minimum-volume box containing the intersection.** We maintain sets of low-complexity by computing the minimum-volume box $\mathcal{B}(\mathbf{b}_{t+1})$ containing the intersection of the initial box $\mathcal{B}(\mathbf{b}_t)$ and the half-space $\mathcal{H}_{\leq}(\mathbf{u}_t)$ obtained from Proposition 4.3 when the preconditioner $\mathbf{P}_t$ fails to yield sufficient decrease.

(b) **Need sufficient backtracking.** If the candidate preconditioner $\mathbf{P}_t$ selected inside the initial box $\mathcal{B}(\mathbf{b}_t)$ is not close enough to 0, there might not be a box smaller than $\mathcal{B}(\mathbf{b}_t)$ that contains the intersection.

Figure 4: **Minimum-volume enclosing boxes**

## 5 Cutting-plane methods

The multidimensional backtracking method is in fact a cutting-plane method that uses separating hyperplanes (from Proposition 4.3) to search for valid preconditioners. The canonical example is the ellipsoid method (Yudin and Nemirovski, 1976; Shor, 1977), but its computational cost is $\Omega(d^2)$ in $\mathbb{R}^d$. We now describe cutting-plane methods with three desirable properties: the preconditioners have good competitive ratios, the feasible set shrinks significantly when backtracking, and the computational cost is $O(d)$. There are many details, but the overall idea is similar to the ellipsoid method.

**A simple warm-up: boxes.** Consider the case when $\mathcal{S}_0$ consists of diagonal matrices with diagonals in the box $\mathcal{B}(\mathbf{b}_0) = \{\,\mathbf{p} \in \mathbb{R}_{\geq 0}^d : \mathbf{p} \leq \mathbf{b}_0\,\}$. We pick a candidate preconditioner by backtracking from the largest point in $\mathcal{B}(\mathbf{b}_0)$ by some constant $\gamma < 1$, taking $\mathbf{P} := \gamma\,\mathrm{Diag}(\mathbf{b}_0)$. If $\mathbf{P}$ satisfies the Armijo condition (4), we take a gradient step. If it does not, we compute the vector $\mathbf{u}_0$ as in Proposition 4.3, and obtain a half-space $\mathcal{H}_{>}(\mathbf{u}_0)$ that contains only invalid preconditioners. We then know we only need to search inside $\mathcal{S}_0 \cap \mathcal{H}_{\leq}(\mathbf{u}_0)$. However, maintaining the set $\mathcal{S}_0 \cap \mathcal{H}_{\leq}(\mathbf{u}_0) \cap \cdots \cap \mathcal{H}_{\leq}(\mathbf{u}_t)$ would be too complex to fit in $O(d)$ time or memory. To reduce complexity, we define $\mathcal{S}_{t+1}$ as the box $\mathcal{B}(\mathbf{b}_{t+1})$ of minimum volume containing $\mathcal{B}(\mathbf{b}_t) \cap \mathcal{H}_{\leq}(\mathbf{u}_t)$, as illustrated in Figure 4a. Due to this restriction, we might not be able to find a smaller set; the original box $\mathcal{B}(\mathbf{b}_t)$ may already be the minimum volume box containing $\mathcal{B}(\mathbf{b}_t) \cap \mathcal{H}_{\leq}(\mathbf{u}_t)$ if $\mathbf{u}_t$ does not cut deep enough, as illustrated in Figure 4b. However, with enough backtracking ($\gamma < 1/d$), we can show that the new box is smaller. This yields the following subroutines to fill in the gaps of Figure 2 (detailed in Appendix D)

$$\text{CANDIDATE}(\mathcal{S}_t, \gamma, \mathbf{x}_t) := \gamma\,\mathrm{Diag}(\mathbf{b}_t), \qquad \text{CUT}(\mathcal{S}_t, \mathbf{P}_t) := \{\,\mathrm{Diag}(\mathbf{p}) : \mathbf{p}_t \in \mathcal{B}(\mathbf{b}_{t+1})\,\}, \qquad (7)$$

where $\mathcal{S}_t = \mathcal{B}(\mathbf{b}_t)$ and $\mathbf{b}_{t+1} := \min\{\mathbf{b}_t, 1/\mathbf{u}_t\}$ element-wise, which give the following guarantees.

**Theorem 5.1.** *Consider the multidimensional backtracking from Figure 2 initialized with a set $\mathcal{S}_0 = \{\,\mathrm{Diag}(\mathbf{p}) : \mathbf{p} \in \mathcal{B}(\mathbf{b}_0)\,\}$ containing $\mathbf{P}_*$, with the subroutines in Equation (7) with $\gamma = 1/2d$. Then: (a) $\mathbf{P}_* \in \mathcal{S}_t$, (b) the candidate preconditioner $\mathbf{P}_t$ is $1/2d$-competitive in $\mathcal{S}_t$ for any $t$, and*

$$(c) \quad \mathrm{Vol}(\mathcal{B}(\mathbf{b}_{t+1})) \leq \frac{1}{d+1}\,\mathrm{Vol}(\mathcal{B}(\mathbf{b}_t)) \quad \text{when } \mathbf{P}_t \text{ fails Equation (4).}$$

*In particular,* CUT *is not called more than* $d\log_{d+1}(L\|\mathbf{b}_0\|_{\infty})$ *times.*

*Proof idea.* To guarantee that the box shrinks, we have to guarantee that the half-space $\mathcal{H}_{\leq}(\mathbf{u}_t)$ cuts deep enough. We know that the half-space has to exclude the query point $\mathbf{P}_t$, i.e. $\langle \mathbf{p}_t, \mathbf{u}_t \rangle \geq 1$, by Proposition 4.2 and that $\mathbf{u}_t \geq 0$ by Proposition 4.3. Querying $\mathbf{P}_t$ sufficiently close to the origin, by taking $\gamma = 1/2d$, is then enough to guarantee the decrease. To bound the total number of cuts, we note that the sets $\mathcal{B}(\mathbf{b}_t)$ have a minimum volume $\mathrm{Vol}_{\min}$, as they have to contain the valid preconditioners. The number of cuts is at most $\log_c(\mathrm{Vol}(\mathcal{B}(b_0))/\mathrm{Vol}_{\min})$ for $c = d+1$. We then bound $\mathrm{Vol}(\mathcal{B}(b_0)) \leq \|\mathbf{b}_0\|_{\infty}^d$ and $\mathrm{Vol}_{\min} \geq 1/L^d$ as $(1/L)\mathbf{I}$ is a valid preconditioner. $\square$

### 5.1 Multidimensional Backtracking with Centered Axis-aligned Ellipsoids

We now improve the competitive ratio from $O(1/d)$ to $O(1/\sqrt{d})$ by switching from boxes to ellipsoids. Whereas general ellipsoids would require $\Omega(d^2)$ complexity (as they involve a $d \times d$ matrix), we consider *centered, axis-aligned ellipsoids*, defined by a diagonal matrix $\mathbf{A} = \mathrm{Diag}(\mathbf{a})$, of the form $\mathcal{E}(\mathbf{a}) := \{\mathbf{p} \in \mathbb{R}_{\geq 0}^d : \|\mathbf{p}\|_{\mathbf{A}} \leq 1\}$, where $\|\mathbf{p}\|_{\mathbf{A}}^2 := \langle \mathbf{p}, \mathbf{A}\mathbf{p} \rangle$. As preconditioners are non-negative, we consider only the positive orthant of the ellipsoid. For simplicity, we refer to those sets as *ellipsoids*.

**Candidate preconditioner.** In the box example, we selected the candidate preconditioner by back-tracking from the largest preconditioner in the box. With an ellipsoid, there is no *largest* preconditioner. We need to choose where to backtrack from. To ensure the candidate preconditioner $\mathbf{P}$ is competitive (Definition 3.1), we backtrack from the preconditioner that maximizes the progress $\|\nabla f(\mathbf{x})\|_{\mathbf{P}}^2$,

$$\underset{\mathbf{p} \in \mathcal{E}(\mathbf{a})}{\arg\max} \|\nabla f(\mathbf{x})\|_{\mathbf{P}}^2 = \frac{\mathbf{A}^{-1}\nabla f(\mathbf{x})^2}{\|\nabla f(\mathbf{x})^2\|_{\mathbf{A}^{-1}}}, \qquad \big(\text{where } \nabla f(\mathbf{x})^2 \coloneqq \nabla f(\mathbf{x}) \odot \nabla f(\mathbf{x})\big). \qquad (8)$$

This lets us pick the preconditioner that makes the most progress *for the current gradient*, and will let us improve the competitive ratio by allowing a backtracking coefficient of $1/\sqrt{d}$ instead of $1/d$.

**Cutting.** To complete the algorithm, we need to find a new set $\mathcal{E}(\mathbf{b}_{t+1})$ with smaller volume which contains the intersection of the previous set $\mathcal{E}(\mathbf{b}_t)$ and the half-space $\mathcal{H}_{\leq}(\mathbf{u}_t)$. Unlike the box approach, the minimum volume ellipsoid has no closed form solution. However, if we backtrack sufficiently, by a factor of $\gamma < 1/\sqrt{d}$, we can find an ellipsoid guaranteed to decrease the volume.

**Lemma 5.2.** *Consider the ellipsoid $\mathcal{E}(\mathbf{a})$ defined by $\mathbf{A} = \mathrm{Diag}(\mathbf{a})$ for $\mathbf{a} \in \mathbb{R}_{>0}^d$. Let $\mathbf{p} \in \mathcal{E}(\mathbf{a})$ be a point sufficiently deep inside the ellipsoid, such that $\|\mathbf{p}\|_{\mathbf{A}} \leq 1/\sqrt{2d}$, and $\mathcal{H}_{>}(\mathbf{u})$ be a half-space obtained from Proposition 4.3 at $\mathbf{p}$. The intersection $\mathcal{E}(\mathbf{a}) \cap \mathcal{H}(\mathbf{u})_{\leq}$ is contained in the new ellipsoid*

$$\mathcal{E}(\mathbf{a}^+(\mathbf{a}, \mathbf{u})), \quad \text{where} \quad \mathbf{a}^+(\mathbf{a}, \mathbf{u}) = \lambda \mathbf{a} + (1-\lambda)\mathbf{u}^2, \quad \lambda = \frac{\ell}{d}\frac{d-1}{\ell-1}, \quad \ell = \|\mathbf{u}\|_{\mathbf{A}^{-1}}^2, \quad (9)$$

*which has a smaller volume, $\mathrm{Vol}(\mathcal{E}(\mathbf{a}^+(\mathbf{a}, \mathbf{u}))) \leq c\, \mathrm{Vol}(\mathcal{E}(\mathbf{a}))$, where $c = \sqrt[4]{e}/\sqrt{2} \approx 0.91$.*

*Proof idea.* The new ellipsoid in (9) is a convex combination between $\mathcal{E}(\mathbf{a})$ and the minimum volume ellipsoid containing the set $\{\mathbf{p} \in \mathbb{R}^d : \langle \mathbf{u}, |\mathbf{p}| \rangle \leq 1\}$ where $|\mathbf{p}|$ is the element-wise absolute value of $\mathbf{p}$. The choice of $\lambda$ in (9) is not optimal, but suffices to guarantee progress as long as $\|\mathbf{p}\|_{\mathbf{A}}$ is small. A similar approach was used by Goemans et al. (2009) to approximate submodular functions, although they consider the polar problem of finding a maximum-volume enclosed ellipsoid. The full proof and discussion on the connections to the polar problem are deferred to Appendix D. $\qquad\square$

To improve the cuts, we can refine the estimate of $\lambda$ in Lemma 5.2 by minimizing the volume numerically. We include this modification, detailed in Appendix D, in our experiments in Section 6.

**Overall guarantees.** We can now define the two subroutines for the ellipsoid method, and obtain the main result that we stated informally in Theorem 1.1, by combining the guarantees of the ellipsoid approach with the convergence result of Proposition 3.2.

**Theorem 5.3.** *Consider the multidimensional backtracking from Figure 2 initialized with the set $\mathcal{S}_0 = \{\mathrm{Diag}(\mathbf{p}) : \mathbf{p} \in \mathcal{E}(\mathbf{a}_0)\}$ containing $\mathbf{P}_*$, given by some scaling $\alpha_0 > 0$ of the uniform vector, $\mathbf{a}_0 = \alpha_0 \mathbf{1}$. For $\mathcal{S}_t$, let $\mathbf{A}_t = \mathrm{Diag}(\mathbf{a}_t)$. Define the subroutines*

$$\mathbf{P}_t = \textsc{candidate}(\mathcal{S}_t, \gamma, \mathbf{x}_t) \coloneqq \gamma \frac{\mathbf{A}_t^{-1}\nabla f(\mathbf{x}_t)^2}{\|\nabla f(\mathbf{x}_t)^2\|_{\mathbf{A}_t^{-1}}}, \quad \textsc{cut}(\mathcal{S}_t, \mathbf{P}_t) \coloneqq \{\mathrm{Diag}(\mathbf{p}) : \mathbf{p} \in \mathcal{E}(\mathbf{a}^+(\mathbf{a}_t, \mathbf{u}_t))\},$$

*where $\mathbf{u}_t$ is the vector given by Proposition 4.3 when $\mathbf{P}_t$ fails the Armijo condition at $\mathbf{x}_t$, and $\mathbf{a}^+$ is computed as in (9). If $\gamma = 1/\sqrt{2d}$, then: (a) $\mathbf{P}_* \in \mathcal{S}_t$ for all $t$, (b) the candidate preconditioners $\mathbf{P}_t$ are $1/\sqrt{2d}$-competitive in $\mathcal{S}_t$, and (c) \textsc{cut} is called no more than $12d\log(L/\alpha_0)$ times.*

# 6 Experiments

To illustrate that multidimensional backtracking finds good preconditioners and improves over gradient descent on ill-conditioned problems even when accounting for the cost of backtracking, we run experiments on small but very ill-conditioned and large ($d \approx 10^6$) problems.

As examples of adaptive gain and hypergradient methods, we include RPROP (Riedmiller and Braun, 1993) and GD with a hypergradient-tuned step-size (GD-HD, Baydin et al. 2018, multiplicative update). As examples of approximate second-order methods, we include diagonal BB (Park et al., 2020) and preconditioned GD using the diagonal of the Hessian. We use default parameters, except for the hypergradient method GD-HD, where we use $10^{-10}$ as the initial step-size instead of $10^{-3}$ to avoid immediate divergence. We include AdaGrad (diagonal), but augment it with a line-search as suggested by Vaswani et al. (2020), to make it competitive in the deterministic setting.

**Line-searches and forward steps.** For all methods that use a line-search, we include a *forward* step, a common heuristic in line-search procedures to allow for larger step-sizes when possible, although it

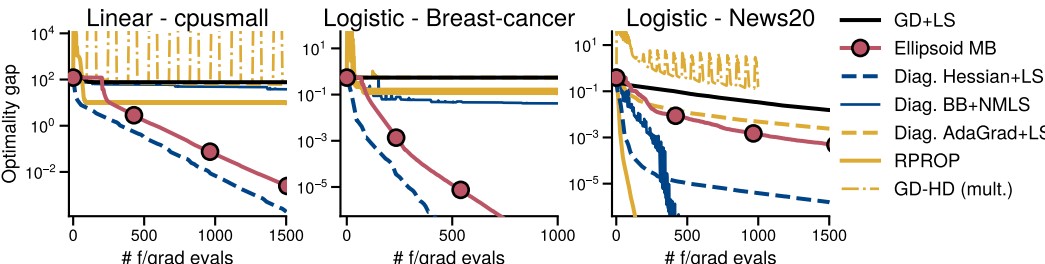

Figure 5: **Multidimensional backtracking finds a good preconditioner, when there is one.** Experiments on regularized linear and logistic regression on small but ill-conditioned datasets, cpusmall and breast-cancer (left, middle), and the large dataset News20 (right, $d \approx 10^6$). Methods used: Gradient Descent (GD) Multidimensional Backtracking (MB) with ellipsoids, diagonal Hessian, diagonal BB, and diagonal AdaGrad—all of which use a line-search (+LS)— RPROP, and GD with hypergradient-tuned step-size (GD-HD) using the multiplicative update. Details in Appendix E.

can increase the number of backtracking steps. When a step-size or preconditioner is accepted, we increase the size of the set, allowing for larger (scalar or per-coordinate) step-sizes by a factor of 1.1.

**Performance comparison.** To capture the overhead of backtracking and provide a fair evaluation, Figures 1 and 5 compare performance per function and gradient evaluations (see Appendix E.1).

**On a small but extremely ill-conditioned problems**, our method is the only one that gets remotely close to being competitive with preconditioning with the diagonal Hessian—while only using first-order information. The diagonal Hessian is very close to the optimal preconditioner for those problems. On the cpusmall dataset, it reduces the condition number from $\kappa \approx 5 \cdot 10^{13}$ to $\approx 300$, while $\kappa_* \approx 150$. All other methods struggle to make progress and stall before a reasonable solution is achieved, indicating they are not competitive with the optimal preconditioner.

**On large regularized logistic regression on News20** ($d \approx 10^6$), gradient descent performs relatively better, suggesting the problem is less ill-conditioned to begin with (the regularized data matrix has condition number $\kappa \approx 10^4$). Despite the bound of $O(d)$ backtracking steps, our methods finds a reasonable preconditioner within 100 gradient evaluations. Despite the high dimensionality, it improves over gradient descent when measured in number of oracle calls.

Using plain gradient updates on the hyperparameters in GD-HD leads to unstable behavior, but diagonal BB and even RPROP, perform remarkably well on some problems—even outperforming preconditioning with the diagonal Hessian, which uses second-order information. However, they fail on other ill-conditioned problems, even when a good diagonal preconditioner exists. This pattern holds across other problems, as shown in Appendix E. Multidimensional backtracking demonstrates robust performance across problems, a clear advantage of having worst-case guarantees.

## 7 Conclusion

We designed *multidimensional backtracking*, an efficient algorithm to automatically find diagonal preconditioners that are competitive with the optimal diagonal preconditioner. Our work provides a definition of adaptive step-sizes that is complementary to the online learning definition. While online learning focuses on the adversarial or highly stochastic setting, we define and show how to find optimal per-coordinate step-sizes in the deterministic smooth convex setting. We show it is possible to build provably robust methods to tune a preconditioner using hypergradients. While our specific implementation uses cutting-planes, the general approach may lead to alternative algorithms, that possibly tune other hyperparameters, with similar guarantees.

The main limitation of our approach is its reliance on the convex deterministic setting. The results might transfer to the stochastic overparametrized regime using the approach of Vaswani et al. (2019), but the non-convex case seems challenging. It is not clear how to get reliable information from a cutting-plane perspective using hypergradients without convexity. As the first method to provably find competitive preconditioners, there are likely modifications that lead to practical improvements while preserving the theoretical guarantees. Possible ideas to improve practical performances include better ways to perform forward steps, using hypergradient information from accepted steps (which are currently ignored), or considering alternative structures to diagonal preconditioners.

## Acknowledgments and Disclosure of Funding

We thank Aaron Mishkin for helpful discussions in the early stages of this work, and Curtis Fox and Si Yi (Cathy) Meng for providing comments on an early version of the manuscript, and for the feedback from anonymous reviewers. This research was partially by the Canada CIFAR AI Chair Program, the Natural Sciences and Engineering Research Council of Canada (NSERC) Discovery Grants RGPIN-2022-03669, and Borealis AI through the Borealis AI Global Fellowship Award.

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

# Supplementary Material

## Contents

Code available at        https://github.com/fKunstner/multidimensional-backtracking

# A Full pseudocode of the algorithms

We first give with a generic version using the subroutines INITIALIZE, CANDIDATE and CUT, to be specialized for the backtracking line-search (Figure 8), multidimensional backtracking using boxes (Figure 10), and ellipsoids (Figure 11). The generic pseudocode is written in terms of preconditioners, but also applies to the step-size version, which we can consider as looking for a preconditioner constrained to isotropic diagonal preconditioners, that is, preconditioners in the set $\{\alpha \mathbf{I} : \alpha \in \mathbb{R}_{\geq 0}\}$.

Although we write the pseudocode maintaining at each iteration an abstract set of preconditioners $\mathcal{S}$, the only information the algorithm needs to maintain on each iteration for the implementation in the different cases is

- **For the line-search:**
  the current maximum step-size $\alpha_{\max}$ defining the interval of valid step-sizes, $[0, \alpha_{\max}]$ such that the set of preconditioners is $\mathcal{S} = \{\alpha \mathbf{I} : \alpha \in [0, \alpha_{\max}]\}$;

- **For multidimensional backtracking with boxes:**
  the vector $\mathbf{b}$ defining the maximum corner of the box $\mathcal{B}(\mathbf{b}) = \{\mathbf{p} \in \mathbb{R}_{\geq 0}^d : \mathbf{p} \leq \mathbf{b}\}$ used to define the candidate diagonals preconditioners in the set $\mathcal{S} = \{\mathrm{Diag}(\mathbf{p}) : \mathbf{p} \in \mathcal{B}(\mathbf{b})\}$;

- **For multidimensional backtracking with ellipsoids:**
  the vector $\mathbf{a}$ defining the axis-aligned ellipsoid $\mathcal{E}(\mathbf{a}) = \{\mathbf{p} \in \mathbb{R}_{\geq 0}^d : \langle \mathbf{p}, \mathrm{Diag}(\mathbf{a})\mathbf{p}\rangle \leq 1\}$ used to define the candidate diagonal preconditioners in the set $\mathcal{S} = \{\mathrm{Diag}(\mathbf{p}) : \mathbf{p} \in \mathcal{E}(\mathbf{a})\}$.

The pseudocode in Figure 6 updates $(\mathbf{x}_t, \mathcal{S}_t)$ to $(\mathbf{x}_{t+1}, \mathcal{S}_{t+1})$ at each iteration, and ensures that either the function value decreases, $f(\mathbf{x}_{t+1}) < f(\mathbf{x}_t)$, or the volume decreases, $\mathrm{Vol}(\mathcal{S}_{t+1}) < \mathrm{Vol}(\mathcal{S}_t)$.

We give an alternative pseudocode in Figure 7, which defines iterations as updates to the iterates $\mathbf{x}_t$ that decrease the function value, and uses a `while`-loop to backtrack. Since it more closely resemble standard ways backtracking line-search is described, some reader may find it easier to understand. We stress, however, that this is still the same algorithm as Figure 6 but written differently.

The pseudocode in Figures 6–11, are expressed in a modular form to highlight how the algorithm works and its similarity to a line-search. In Appendix A.5, we give a more directly implementable pseudocode of multidimensional backtracking in both box and ellipsoid variants solely relying on vector notation.

Figure 6: **Generic pseudocode for the line-search or multidimensional backtracking.** uses the subroutines INITIALIZE, CANDIDATE, CUT defined in the later sections.

---

**Backtracking Preconditioner Search with Sets**

**Input:**
    A starting point $\mathbf{x}_0 \in \mathbb{R}^d$;
    A backtracking coefficient $\gamma \in [0, 1]$;
    A scalar $c_0 > 0$ larger than the optimal preconditioner, i.e., such that $\mathbf{P}_* \preceq c_0 \mathbf{I}$.

$\mathcal{S}_0 = \text{INITIALIZE}(c_0)$

Iterate for $t$ in $0, 1, ..., T-1$
    $\mathbf{P}_t = \text{CANDIDATE}(\mathcal{S}_t, \gamma, \nabla f(\mathbf{x}_t))$
    If $f(\mathbf{x}_t - \mathbf{P}_t \nabla f(\mathbf{x}_t)) \leq f(\mathbf{x}_t) - \frac{1}{2}\|\nabla f(\mathbf{x}_t)\|_{\mathbf{P}_t}^2$         // Armijo condition Equation (4)
        $(\mathbf{x}_{t+1}, \mathcal{S}_{t+1}) = (\mathbf{x}_t - \mathbf{P}_t \nabla f(\mathbf{x}_t), \mathcal{S}_t)$
    Otherwise,
        $(\mathbf{x}_{t+1}, \mathcal{S}_{t+1}) = (\mathbf{x}_t, \text{CUT}(\mathcal{S}_t, \mathbf{x}_t, \mathbf{P}_t))$

**Output:** $\mathbf{x}_T$

---

Figure 7: **Alternative pseudocode for the line-search or multidimensional backtracking.** Uses a while-loop for backtracking and only updates the iterates $\mathbf{x}_t$ when they lead to progress.

---

**Backtracking Preconditioner Search with Sets – `while`-loop variant**

**Input:**
    A starting point $\mathbf{x}_0 \in \mathbb{R}^d$;
    A backtracking coefficient $\gamma \in [0, 1]$;
    A scalar $c_0 > 0$ larger than the best preconditioner, that is, $\mathbf{P}_* \preceq c_0 \mathbf{I}$.

Initialize the set $\mathcal{S} = \text{INITIALIZE}(c_0)$
Iterate for $t$ in $0, 1, ..., T-1$
    $\mathbf{P}_t \leftarrow \text{CANDIDATE}(\mathcal{S}_b, \gamma, \nabla f(\mathbf{x}_t))$
    While $f(\mathbf{x}_t - \mathbf{P}_t \nabla f(\mathbf{x}_t)) \leq f(\mathbf{x}_t) - \frac{1}{2}\|\nabla f(\mathbf{x}_t)\|_{\mathbf{P}_t}^2$         // Armijo condition Equation (4)
        $\mathcal{S} \leftarrow \text{CUT}(\mathcal{S}, \mathbf{x}_t, \mathbf{P}_t)$
        $\mathbf{P}_t \leftarrow \text{CANDIDATE}(\mathcal{S}, \gamma, \nabla f(\mathbf{x}_t))$
    $\mathbf{x}_{t+1} = \mathbf{x}_t - \mathbf{P}_t \nabla f(\mathbf{x}_t)$

**Output:** $\mathbf{x}_T$

---

## A.1   Subroutines for standard backtracking line-search

Implementation of the subroutines for the standard backtracking line-search. Although written in terms of sets, the algorithm only needs to maintain the maximum step-size in the interval $[0, \alpha_{\max}]$ at each iteration. The corresponding preconditioners are the matrices $\mathcal{S} = \{\alpha \mathbf{I} : \alpha \in [0, \alpha_{\max}]\}$.

Figure 8: **Specialization of the subroutines for the backtracking line-search**

---

INITIALIZE($c_0$)

**Input:**
    A scaling $c_0$ larger than the optimum step-size, that is, such that $1/L \leq c_0$.

**Output:** Set of preconditioners $\mathcal{S} = \{\alpha \mathbf{I} : \alpha \in [0, c_0]\}$.

---

CANDIDATE($\mathcal{S}, \gamma, \mathbf{x}$)

**Input:**
    Set of scalar preconditioners $\mathcal{S} = \{\alpha \mathbf{I} : \alpha \in [0, \alpha_{\max}]\}$;
    Backtracking coefficient $\gamma \in [0, 1]$;
    Current iterate $\mathbf{x} \in \mathbb{R}^d$.             //Not used for the step-size version.

**Output:** Preconditioner $\gamma \alpha_{\max} \mathbf{I}$

---

CUT($\mathcal{S}, \mathbf{x}, \mathbf{P}$)

**Input:**
    Set of scalar preconditioners $\mathcal{S} = \{\alpha \mathbf{I} : \alpha \in [0, \alpha_{\max}]\}$;
    Current iterate $\mathbf{x} \in \mathbb{R}^d$;             //Not used for the step-size version.
    Preconditioner $\mathbf{P} = \alpha_{\mathrm{bad}} \mathbf{I}$ that failed the Armijo condition at $\mathbf{x}$.

**Output:** Set of scalar preconditioners with reduced interval, $\mathcal{S} = \{\alpha \mathbf{I} : \alpha \in [0, \alpha_{\mathrm{bad}}]\}$

---

## A.2   Separating hyperplanes used by multidimensional backtracking

Both versions of multidimensional backtracking need a direction to update the set of preconditioners in the CUT subroutine. We define the subroutine SEPARATINGHYPERPLANE in Figure 9. The description of the separating hyperplane and their properties can be found in Section 4 and Appendix C.

Figure 9: **Separating hyperplane used by both variants of multidimensional backtracking.**

---

SEPARATINGHYPERPLANE($\mathbf{x}, \mathbf{P}$) for diagonal preconditioners

**Input:**
    Current iterate $\mathbf{x} \in \mathbb{R}^d$;
    Diagonal preconditioner $\mathbf{P} = \mathrm{Diag}(\mathbf{p})$ that failed the Armijo condition at $\mathbf{x}$.

$\mathbf{x}^+ = \mathbf{x} - \mathbf{P}\nabla f(\mathbf{x})$
$\mathbf{g}\ \ = \nabla f(\mathbf{x})$
$\mathbf{g}^+ = \nabla f(\mathbf{x}^+)$

$\mathbf{v} = \dfrac{\left(\frac{1}{2}\mathbf{g} - \mathbf{g}^+\right) \odot \mathbf{g}}{f(\mathbf{x}) - f(\mathbf{x}^+) - \langle \mathbf{g}, \mathbf{P}\mathbf{g}^+ \rangle}$        //Separating hyperplane from Proposition 4.2

**Output:** $\mathbf{u} = \max\{\mathbf{v}, 0\}$ element-wise        //Stronger hyperplane from Proposition 4.3

---

### A.3 Multidimensional backtracking using boxes

The implementation of multidimensional backtracking with boxes only needs to maintain a vector $\mathbf{b}$, representing the maximum step-size for each coordinate that has not been ruled out, in the box $\mathcal{B}(\mathbf{b})$. The associated sets of preconditioners $\mathcal{S}$ are

$$\mathcal{B}(\mathbf{b}) = \{\mathbf{p} \in \mathbb{R}_{\geq 0}^d : \mathbf{p} \leq \mathbf{b}\}, \qquad \mathcal{S} = \{\mathrm{Diag}(\mathbf{p}) : \mathbf{p} \in \mathcal{B}(\mathbf{b})\}.$$

The description of boxes and the theoretical guarantees when using them in multidimensional backtracking can be found in Section 5 and Appendix D.1. The subroutines used by the algorithm with boxes are:

- INITIALIZE: initializes $\mathbf{b}$ to $c_0 \mathbf{1}$ so that the diagonal preconditioner $c_0 \mathbf{I}$ is in $\mathcal{S}_0$.

- CANDIDATE: backtracks from the largest diagonal in $\mathcal{B}(\mathbf{b})$, returning $\gamma \, \mathrm{Diag}(\mathbf{b})$.

- SEPARATINGHYPERPLANE: computes the vector $\mathbf{u}$ defining the half-space of invalid preconditioners $\mathcal{H}_{>}(\mathbf{u})$ obtained when the preconditioner $\mathbf{P}$ fails the Armijo condition at $\mathbf{x}$ as described in Proposition 4.2 and Proposition 4.3.

- CUT: returns the minimum volume box $\mathcal{B}(\mathbf{b}^+)$ containing the intersection $\mathcal{B}(\mathbf{b}) \cap \mathcal{H}_{\leq}(\mathbf{u})$.

Figure 10: **Specialization of the subroutines for multidimensional backtracking with boxes**

---

INITIALIZE$(c_0)$

**Input:**
 A scalar $c_0$ such that $c_0 \mathbf{I}$ is larger than the optimal diagonal preconditioner, i.e., $\mathbf{P}_* \preceq c_0 \mathbf{I}$.

**Output:** Set $\mathcal{S} = \{\mathrm{Diag}(\mathbf{p}) : \mathbf{p} \in \mathcal{B}(\mathbf{b})\}$ with $\mathcal{B}(\mathbf{b}) = \{\mathbf{p} \in \mathbb{R}_{\geq 0}^d : \mathbf{p} \leq \mathbf{b}\}$ where $\mathbf{b} := c_0 \mathbf{1}$

---

CANDIDATE$(\mathcal{S}, \gamma, \mathbf{x})$

**Input:**
 Set of preconditioners $\mathcal{S} = \{\mathrm{Diag}(\mathbf{p}) : \mathbf{p} \in \mathcal{B}(\mathbf{b})\}$ with $\mathcal{B}(\mathbf{b}) = \{\mathbf{p} \in \mathbb{R}_{\geq 0}^d : \mathbf{p} \leq \mathbf{b}\}$;
 Backtracking coefficient $\gamma \in [0, 1]$;
 Current iterate $\mathbf{x} \in \mathbb{R}^d$.     //Not used for the box version.

**Output:** Preconditioner $\gamma \, \mathrm{Diag}(\mathbf{b})$

---

CUT$(\mathcal{S}, \mathbf{x}, \mathbf{P})$

**Input:**
 Set of preconditioners $\mathcal{S} = \{\mathrm{Diag}(\mathbf{p}) : \mathbf{p} \in \mathcal{B}(\mathbf{b})\}$ with $\mathcal{B}(\mathbf{b}) = \{\mathbf{p} \in \mathbb{R}_{\geq 0}^d : \mathbf{p} \leq \mathbf{b}\}$;
 Backtracking coefficient $\gamma \in [0, 1]$;
 Preconditioner $\mathbf{P}_{\mathrm{bad}}$ that failed the Armijo condition at $\mathbf{x}$.

$\mathbf{u} = \mathrm{SEPARATINGHYPERPLANE}(\mathbf{x}_t, \mathbf{P}_{\mathrm{bad}})$
$\mathbf{b}^+ = \max\{\mathbf{b}, 1/\mathbf{u}\}$     //Minimum volume box $\mathcal{B}(\mathbf{b}^+)$ containing $\mathcal{B}(\mathbf{b}) \cap \mathcal{H}_{\leq}(\mathbf{u})$

**Output:** Set of diagonal preconditioners $\mathcal{S} = \{\mathrm{Diag}(\mathbf{p}) : \mathbf{p} \in \mathcal{B}(\mathbf{b}^+)\}$.

---

### A.4 Multidimensional backtracking using ellipsoids

The implementation only needs to maintain a vector $\mathbf{a}$ representing the diagonal of the matrix defining the (centered, axis-alligned) ellipsoid $\mathcal{E}(\mathbf{a})$ and the associated set of preconditioners $\mathcal{S}$ given by

$$\mathcal{E}(\mathbf{a}) = \{\mathbf{p} \in \mathbb{R}^d_{\geq 0} : \langle \mathbf{p}, \mathrm{Diag}(\mathbf{a})\mathbf{p} \rangle \leq 1\}, \qquad \mathcal{S} = \{\mathrm{Diag}(\mathbf{p}) : \mathbf{p} \in \mathcal{E}(\mathbf{a})\}.$$

The description of the ellipsoids and their properties can be found in Section 5 and Appendix D.2. The subroutines used by the algorithm with boxes are:

- INITIALIZE: initializes $\mathbf{a}$ to $(1/dc_0^2)\mathbf{1}$ so that $c_0\mathbf{1} \in \mathcal{E}(\mathbf{a})$, implying the diagonal preconditioner $c_0\mathbf{I}$ is in $\mathcal{S}$.

- CANDIDATE: backtracks from the diagonal preconditioner in $\mathcal{S}$ that maximizes the gradient norm. Let $\mathcal{E}(\mathbf{a})$ be the set of candidate diagonals and define $\mathbf{A} = \mathrm{Diag}(\mathbf{a})$. The subroutine returns $\gamma\mathbf{P}_{\max}$, where

$$\mathbf{P}_{\max} := \arg\max_{\mathbf{P} \in \mathcal{S}} \|\nabla f(\mathbf{x})\|^2_{\mathbf{P}}.$$

  Writing this in terms of the diagonal vector $\mathbf{p}_{\max} := \mathrm{diag}(\mathbf{P}_{\max})$ yields

$$\mathbf{p}_{\max} = \arg\max_{\mathbf{p} \in \mathcal{E}(\mathbf{a})} \|\nabla f(\mathbf{x})\|^2_{\mathrm{Diag}(\mathbf{p})},$$

$$= \arg\max_{\mathbf{p}} \langle \nabla f(\mathbf{x})^2, \mathbf{p} \rangle : \|\mathbf{p}\|_{\mathbf{A}} \leq 1 = \frac{\mathbf{A}^{-1}\nabla f(\mathbf{x})^2}{\|\nabla f(\mathbf{x})\|_{\mathbf{A}^{-1}}},$$

  where $\nabla f(\mathbf{x})^2 = \nabla f(\mathbf{x}) \odot \nabla f(\mathbf{x})$.

- SEPARATINGHYPERPLANE: computes the vector $\mathbf{u}$ defining the half-space of invalid preconditioners $\mathcal{H}_{>}(\mathbf{u})$ obtained when the preconditioner $\mathbf{P}$ fails the Armijo condition at $\mathbf{x}$ as described in Proposition 4.2 and Proposition 4.3.

- CUT: returns an ellipsoid $\mathcal{E}(\mathbf{a}^+)$ containing the intersection of $\mathcal{E}(\mathbf{a}) \cap \mathcal{H}_{\leq}(\mathbf{u})$ with guaranteed volume decrease from $\mathcal{E}(\mathbf{a})$. As there is no closed-form solution for the minimum volume ellipsoid, we set $\mathbf{a}^+$ as a convex combination between the original ellipsoid $\mathcal{E}(\mathbf{a})$ and the minimum volume axis-aligned ellipsoid containing $\mathcal{H}_{\leq}(\mathbf{u})$, given by $\mathcal{E}(\mathbf{u}^2)$, that is,

$$\mathbf{a}^+ := \lambda\mathbf{a} + (1-\lambda)\mathbf{u}^2, \quad \text{where} \quad \lambda := \frac{\ell}{d}\frac{d-1}{\ell-1} \quad \text{and} \quad \ell := \|\mathbf{u}\|^2_{\mathbf{A}^{-1}}, \quad (10)$$

  where $\mathbf{A} := \mathrm{diag}(\mathbf{a})$. Although the above choice of $\lambda$ has guaranteed volume decrease, we can find a better value of $\lambda$ by solving the minimum volume ellipsoid as a function of $\lambda$ numerically. Namely, approximating

$$\lambda^* := \arg\min_{0 < \lambda < 1} -\log(\det(\lambda\,\mathrm{Diag}(\mathbf{a}) + (1-\lambda)\,\mathrm{Diag}(\mathbf{u}^2))).'$$

In our experiments, we start with $\lambda$ as in (10) and, starting from it, we solve the above minimization problem numerically using L-BFGS-B (Zhu et al., 1997) in SciPy (Virtanen et al., 2020). This preserves the theoretical guarantee while improving empirical performance.

Figure 11: **Specialization of the subroutines for multidimensional backtracking with ellipsoids**

---

INITIALIZE($c_0$)

**Input:**
A scalar $c_0 > 0$ such that $c_0\mathbf{I}$ is larger than the optimal diagonal preconditioner, i.e., $\mathbf{P}_* \preceq c_0\mathbf{I}$.

**Output:** $\mathcal{S} = \{\text{Diag}(\mathbf{p}) : \mathbf{p} \in \mathcal{E}(\mathbf{a})\}$ with $\mathcal{E}(\mathbf{a}) = \{\mathbf{p} \in \mathbb{R}^d_{\geq 0} : \langle \mathbf{p}, \text{Diag}(\mathbf{a})\mathbf{p}\rangle \leq 1\}$ for $\mathbf{a} = \frac{1}{dc_0^2}\mathbf{1}$

---

CANDIDATE($\mathcal{S}, \gamma, \mathbf{x}$)

**Input:**
A set $\mathcal{S} = \{\text{Diag}(\mathbf{p}) : \mathbf{p} \in \mathcal{E}(\mathbf{a})\}$ where $\mathcal{E}(\mathbf{a}) = \{\mathbf{p} \in \mathbb{R}^d_{\geq 0} : \langle \mathbf{p}, \text{Diag}(\mathbf{a})\mathbf{p}\rangle \leq 1\}$, and $\mathbf{a} \in \mathbb{R}^d_{>0}$;
Backtracking coefficient $\gamma \in [0, 1]$;
Current iterate $\mathbf{x} \in \mathbb{R}^d$.

$\mathbf{d} = \nabla f(\mathbf{x}) \circ \nabla f(\mathbf{x})$
$\mathbf{p}_{\max} = \dfrac{\mathbf{A}^{-1}\mathbf{d}}{\|\mathbf{d}\|_{\mathbf{A}^{-1}}}$ $\hspace{3cm}$ //Where $\mathbf{A} = \text{Diag}(\mathbf{a})$

**Output:** Preconditioner $\gamma\,\text{Diag}(\mathbf{p}_{\max})$

---

CUT($\mathcal{S}, \mathbf{x}, \mathbf{P}$)

**Input:**
A set $\mathcal{S} = \{\text{Diag}(\mathbf{p}) : \mathbf{p} \in \mathcal{E}(\mathbf{a})\}$, where $\mathcal{E}(\mathbf{a}) = \{\mathbf{p} \in \mathbb{R}^d_{\geq 0} : \langle \mathbf{p}, \text{Diag}(\mathbf{a})\mathbf{p}\rangle \leq 1\}$, and $\mathbf{a} \in \mathbb{R}^d_{>0}$;
Current iterate $\mathbf{x} \in \mathbb{R}^d$;
Preconditioner $\mathbf{P}_{\text{bad}}$ that failed the Armijo condition at $\mathbf{x}$.

$\mathbf{u} = \text{SEPARATINGHYPERPLANE}(\mathbf{x}_t, \mathbf{P}_{\text{bad}})$
$\ell = \|\mathbf{u}\|^2_{\mathbf{A}^{-1}}$
$\lambda = \frac{\ell}{d}\frac{d-1}{\ell-1}$ $\hspace{0.3cm}$ $\big($or numerically solve $\lambda = \arg\min_{0<c<1} -\log(\det(c\,\text{Diag}(\mathbf{a}) + (1-c)\,\text{Diag}(\mathbf{u}^2)))\big)$
$\mathbf{a}^+ = \lambda\mathbf{a} + (1-\lambda)\mathbf{u}^2$ $\hspace{1.5cm}$ //Approx. min. volume ellipsoid $\mathcal{E}(\mathbf{a}^+)$ containing $\mathcal{E}(\mathbf{a}) \cap \mathcal{H}_{\leq}(\mathbf{u})$

**Output:** The set $\mathcal{S} = \{\text{Diag}(\mathbf{p}) : \mathbf{p} \in \mathcal{E}(\mathbf{a}^+)\}$

---

## A.5 Implementable pseudocode

The pseudocode in Figures 6–11 are expressed in a modular form to highlight how the algorithm works and its similarity to a line-search. In this section, we give a more directly implementable pseudocode of multidimensional backtracking, in both the box and ellipsoid variants, using mostly vector notation. Scalar operations on vectors such as $\mathbf{u}/\mathbf{a}$, $\sqrt{\mathbf{u}}$, $\mathbf{u}^2$ are understood to be taken element-wise.

---

**Multidimensional backtracking using boxes**                  Direct implementation

**Input:**

     Function to optimize $f \colon \mathbb{R}^d \to \mathbb{R}$ ;

     Starting point $\mathbf{x}_0 \in \mathbb{R}^d$;

     A scalar for the scale of initial set of preconditioners $c_0 > 0$;

     Backtracking coefficient $\gamma < 1/d$.

$\mathbf{b} = c_0 \mathbf{1}$                                           //Initialize box

Iterate for $t$ in $0, 1, ...$

     $\mathbf{p}_t = \gamma \mathbf{b}$                                   //Get candidate preconditioner

     $\mathbf{g}_t = \nabla f(\mathbf{x}_t)$                              //Get candidate point

     $\mathbf{x}_t^+ = \mathbf{x}_t - \mathbf{p}_t \circ \mathbf{g}_t$

     While $f(\mathbf{x}_t^+) > f(\mathbf{x}_t) - \frac{1}{2}\langle \mathbf{g}_t^2, \mathbf{p}_t \rangle$              //Armijo condition fails

         $\mathbf{g}_t^+ = \nabla f(\mathbf{x}_t^+)$                    //Get next gradient to compute

         $\mathbf{d}_t = (\frac{1}{2}\mathbf{g}_t - \mathbf{g}_t^+) \circ \mathbf{g}_t$             //the separating hyperplane direction,

         $c_t = f(\mathbf{x}_t) - f(\mathbf{x}_t^+) - \langle \mathbf{g}_t \circ \mathbf{p}_t, \mathbf{g}_t^+ \rangle$     //the normalization constant,

         $\mathbf{u}_t = \max\{\mathbf{d}_t/c_t, 0\}$ (element-wise)         //and truncate it

         $\mathbf{b} = \mathbf{1}/\max\{\mathbf{1}/\mathbf{b}, \mathbf{u}_t\}$ (element-wise)      //Find new minimum volume box.

                                                   //($\infty$-free $\min\{\mathbf{b}, 1/\mathbf{u}\}$)

         $\mathbf{p}_t = \gamma \mathbf{b}$                             //Pick next candidate preconditioner

         $\mathbf{x}_t^+ = \mathbf{x}_t - \mathbf{p}_t \circ \mathbf{g}_t$                   //and next candidate point

     $\mathbf{x}_{t+1} = \mathbf{x}_t^+$                                  //Accept new point

**Output:** $\mathbf{x}_t$

---

| **Multidimensional backtracking using ellipsoids** | Direct implementation |

**Input:**

    Function to optimize $f\colon \mathbb{R}^d \to \mathbb{R}$ ;
    Starting point $\mathbf{x}_0 \in \mathbb{R}^d$;
    A scalar for the scale of initial set of preconditioners $c_0 > 0$;
    Backtracking coefficient $\gamma < {}^1\!/\!\sqrt{d}$

$\mathbf{a} = \mathbf{1}/(dc_0^2)$               //Initialize ellipsoid

Iterate for $t$ in $0, 1, ...$

    $\mathbf{g}_t = \nabla f(\mathbf{x}_t)$
    $\mathbf{p}_t = \mathbf{g}_t^2/\mathbf{a}$ (element-wise)     //Get candidate preconditioner
    $\mathbf{p}_t = \gamma\mathbf{p}_t/\big\|\mathbf{g}_t^2/\sqrt{\mathbf{a}}\big\|$ (element-wise)     //normalize it

    $\mathbf{x}_t^+ = \mathbf{x}_t - \mathbf{p}_t \circ \mathbf{g}_t$     //Get candidate point

    While $f(\mathbf{x}_t^+) > f(\mathbf{x}_t) - \frac{1}{2}\big\langle\mathbf{g}_t^2, \mathbf{p}_t\big\rangle$     //Armijo condition fails

        $\mathbf{g}_t^+ = \nabla f(\mathbf{x}_t^+)$     //Get next gradient to compute
        $\mathbf{d}_t = (\frac{1}{2}\mathbf{g}_t - \mathbf{g}_t^+) \circ \mathbf{g}_t$     //the separating hyperplane direction,
        $c_t = f(\mathbf{x}_t) - f(\mathbf{x}_t^+) - \big\langle\mathbf{g}_t \circ \mathbf{p}_t, \mathbf{g}_t^+\big\rangle$     //the normalization constant,
        $\mathbf{u}_t = \max\{\mathbf{d}_t/c_t, 0\}$ (element-wise)     //and truncate it

        take $\lambda = \frac{\ell(d-1)}{d(\ell-1)}$ where $\ell = \big\langle\mathbf{u}^2, 1/\mathbf{a}\big\rangle$     //Approx. min. vol. new ellipsoid
        or     //
        find $\lambda$ by numerically minimizing $\phi(\lambda)$ where     //Find better approximation of min.
            $\phi(\lambda) = -\sum_{i=1}^d \log(\lambda\mathbf{a}[i] + (1-\lambda)\mathbf{u}_t[i]^2)$     //of volume of new ellipsoid

        $\mathbf{a} = \lambda\mathbf{a} + (1-\lambda)\mathbf{u}^2$     //New ellipsoid

        $\mathbf{p}_t = \mathbf{g}_t^2/\mathbf{a}$ (element-wise)     //Get new candidate preconditioner,
        $\mathbf{p}_t = \gamma\mathbf{p}_t/\big\|\mathbf{g}_t^2/\sqrt{\mathbf{a}}\big\|$ (element-wise)     //normalized,
        $\mathbf{x}_t^+ = \mathbf{x}_t - \mathbf{p}_t \circ \mathbf{g}_t$     //and new candidate point

    $\mathbf{x}_{t+1} = \mathbf{x}_t^+$     //Accept new point

**Output:** $\mathbf{x}_t$

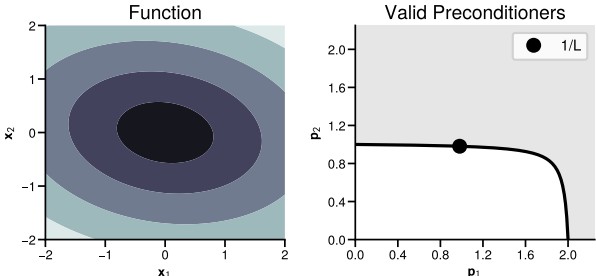

Figure 12: Set of valid diagonal preconditioners (step-sizes $\mathbf{p}_1$ and $\mathbf{p}_2$) for the quadratic in Equation (11). Preconditioned gradient descent can use a larger step-size in the first coordinate.

## B   Optimal preconditioners, valid preconditioners and competitive ratios

In Section 2, we defined the optimal preconditioner $\mathbf{P}_*$ as the preconditioner that is the best overall approximation to the inverse Hessian. Formally, we define the optimal diagonal preconditioner $\mathbf{P}_*$ as

$$\mathbf{P}_* := \operatorname*{arg\,min}_{\mathbf{P}\succ 0,\text{diagonal}} \kappa \quad \text{such that} \quad \frac{1}{\kappa}\mathbf{P}^{-1} \preceq \nabla^2 f(\mathbf{x}) \preceq \mathbf{P}^{-1} \text{ for all } \mathbf{x}. \tag{1}$$

One way to interpret this definition is that $\mathbf{P}_*^{-1}$ is the tightest diagonal approximation to $\nabla^2 f(\mathbf{x})$.

We remark that we do not need $f$ to be (strongly-)convex to define the theoretically optimal step-size of $1/L$ for gradient descent. Thus, one may wonder why we need strong-convexity (although we relax this to requiring $f$ to be PL in Appendix B.1) to define what an optimal preconditioner is in (1).

The main difference between the scalar step-size and per-coordinate step-sizes settings is whether the "largest" step-size or preconditioner is well-defined. In the scalar setting, the largest step-size that is guaranteed to lead to progress everywhere (i.e., a step-size that satisfies the Armijo condition (3) for all $\mathbf{x}$) is well-defined and equal to $\alpha_* := 1/L$ for $L$-smooth function $f$. Equivalently,

$$\alpha_* = \sup\left\{\alpha > 0 : \nabla^2 f(\mathbf{x}) \preceq \frac{1}{\alpha}\mathbf{I}\right\} = \sup_{\mathbf{x}\in\mathbb{R}^d} \lambda_{\max}(\nabla^2 f(\mathbf{x})),$$

where $\lambda_{\max}(\nabla^2 f(\mathbf{x}))$ is the largest eigenvalue of $\nabla^2 f(\mathbf{x})$. But in the case of preconditioners, the ordering on positive definite matrices is not complete, so there is no single "largest" preconditioner $\mathbf{P}$ that satisfies $\nabla^2 f(\mathbf{x}) \preceq \mathbf{P}^{-1}$. We can still describe "good" preconditioners, that are guaranteed to satisfy the Armijo condition (Equation (4)) everywhere; this is the notion of valid preconditioners defined in Definition 4.1, which in set notation is $\mathcal{V} := \{\mathbf{P} \succ 0 : \nabla^2 f(\mathbf{x}) \preceq \mathbf{P}^{-1}\}$. With this definition, we can consider the set of valid preconditioners $\mathbf{P}$ for which there are no bigger valid preconditioners, that is, $\mathcal{P} := \{\mathbf{P} \in \mathcal{V} : \nexists \mathbf{P}' \in \mathcal{V} \text{ s.t. } \mathbf{P} \prec \mathbf{P}'\}$. However, $\mathcal{P}$ contains incomparable preconditioners, that is, distinct matrices $\mathbf{A}, \mathbf{B} \in \mathcal{P}$ that neither $\mathbf{A} \succeq \mathbf{B}$ nor $\mathbf{A} \preceq \mathbf{B}$ hold.

Let us look at an example with a quadratic function (illustrated in Figure 12)

$$f(\mathbf{x}) = \frac{1}{2}\langle \mathbf{x}, \mathbf{A}\mathbf{x}\rangle \qquad \text{with Hessian} \qquad \mathbf{A} = \begin{bmatrix} .5 & .1 \\ .1 & 1.0 \end{bmatrix}. \tag{11}$$

There are many preconditioners that are valid,[3] for example using the per-coordinate step-sizes

$$\mathbf{P}_L \approx \begin{bmatrix} .91 & 0 \\ 0 & .91 \end{bmatrix}, \qquad \mathbf{P}_1 = \begin{bmatrix} 2.0 & 0 \\ 0 & 0.0 \end{bmatrix}, \qquad \mathbf{P}_2 = \begin{bmatrix} 0.0 & 0 \\ 0 & 1.0 \end{bmatrix}, \qquad \mathbf{P}_* \approx \begin{bmatrix} 1.75 & 0 \\ 0 & 0.87 \end{bmatrix}.$$

The preconditioner $\mathbf{P}_L$ corresponds to the $1/L$ step-size, $\mathbf{P}_1$ and $\mathbf{P}_2$ take the largest possible step-size in each coordinate, and $\mathbf{P}_*$ is the optimal preconditioner according to Equation (1). Those preconditioners are not comparable to each other, as neither $\mathbf{P}_L \prec \mathbf{P}_*$ nor $\mathbf{P}_* \prec \mathbf{P}_L$ hold. Instead of looking at the matrices themselves, we use in (1) the condition number[4] of $\mathbf{P}^{1/2}\nabla^2 f(\mathbf{x})\mathbf{P}^{1/2}$ as a measure of quality of $\mathbf{P}$. This allows for a well-defined optimal preconditioner as this condition number can be maximized.

---

[3]Up to invertibility issues which we address in the next subsection.

[4]Our definition is slightly different, but both notions are equivalent for positive definite $\mathbf{P}$.

## B.1 Defining optimal preconditioners without twice-differentiability or strong-convexity

Although we used twice-differentiability of $f$ to define the optimal preconditioner, this is not necessary. If $f$ is not twice-differentiable but still strongly-convex, the definition in Equation (1) can be replaced by Equation (2), as finding the $\mathbf{P}$-norm under which the function is most strongly-convex.

$$\mathbf{P}_* = \operatorname*{arg\,min}_{\mathbf{P} \succ 0,\ \text{diagonal}} \kappa$$

$$\text{such that } \begin{cases} \frac{1}{\kappa}\frac{1}{2}\|\mathbf{x}-\mathbf{y}\|_{\mathbf{P}^{-1}}^2 \leq f(\mathbf{y}) - f(\mathbf{x}) - \langle \nabla f(\mathbf{x}), \mathbf{y}-\mathbf{x}\rangle, \\ f(\mathbf{y}) - f(\mathbf{x}) - \langle \nabla f(\mathbf{x}), \mathbf{y}-\mathbf{x}\rangle \leq \frac{1}{2}\|\mathbf{y}-\mathbf{x}\|_{\mathbf{P}^{-1}}^2, \end{cases} \quad \text{for all } \mathbf{x}, \mathbf{y}.$$

To avoid strong-convexity, we can instead use the PL inequality. A function $f$ is $\mu$-PL if

$$\frac{1}{\mu}\frac{1}{2}\|\nabla f(\mathbf{x})\|^2 \geq f(\mathbf{x}) - f(\mathbf{x}_*). \tag{12}$$

This property is implied by $\mu$-strong convexity. We refer to the work of Karimi et al. (2016) for the properties of PL functions and its relation to other assumptions. To adapt Equation (12) to our results, we can measure the PL constant $\mu$ in the norm induced by $\mathbf{P}$, and say that $f$ is $\mu$-PL in $\|\cdot\|_{\mathbf{P}}$ if

$$\frac{1}{\mu}\frac{1}{2}\|\nabla f(\mathbf{x})\|_{\mathbf{P}}^2 \geq f(\mathbf{x}) - f(\mathbf{x}_*). \tag{13}$$

We use this inequality in the convergence proof in Proposition 3.2 since it is a consequence of strong-convexity. As this property is the only property of strong-convexity needed for our results, we can adapt our results to be competitive with the optimal preconditioner defined using the PL inequality, using the definition

$$\mathbf{P}_*^{\text{PL}} := \operatorname*{arg\,min}_{\mathbf{P} \succ 0,\ \text{diagonal}} \kappa$$

$$\text{such that } \begin{cases} \frac{1}{\kappa}\|\nabla f(\mathbf{x})\|_{\mathbf{P}}^2 \geq f(\mathbf{x}) - f(\mathbf{x}_*) & \text{for all } \mathbf{x}, \\ f(\mathbf{y}) - f(\mathbf{x}) - \langle \nabla f(\mathbf{x}), \mathbf{y}-\mathbf{x}\rangle \leq \frac{1}{2}\|\mathbf{y}-\mathbf{x}\|_{\mathbf{P}^{-1}}^2, & \text{for all } \mathbf{x}, \mathbf{y}. \end{cases} \tag{14}$$

If $f$ is $\mu$-PL and $L$-smooth, Equation (14) has a feasible solution at $\mathbf{P} = 1/L\mathbf{I}$ number $\kappa = L/\mu$. The constraint based on the $\mu$-PL condition in Equation (14) is weaker than the definition using strong-convexity, as strong-convexity implies the PL inequality. The optimal preconditioner defined using the PL inequality (14) might thus achieve a lower condition number than the one using strong-convexity (1). For example, the quadratic $f(\mathbf{x}) = (1/2)\langle \mathbf{x}, \mathbf{A}\mathbf{x}\rangle$ with a positive semi-definite $\mathbf{A}$ is not strongly convex if the smallest eigenvalue of $\mathbf{A}$ is 0. The optimal preconditioner in Equation (1) is ill-defined (or has condition number $\kappa_* = \infty$). In contrast, the optimal preconditioner defined using the PL inequality in Equation (14) has a finite condition number, as $\mathbf{P} = 1/L\mathbf{I}$ is a feasible solution with condition number $\kappa = L/\lambda_{\min}^+(\mathbf{A})$ where $\lambda_{\min}^+(\mathbf{A})$ is the smallest non-zero eigenvalue of $\mathbf{A}$. As our proofs only use the properties guaranteed by Equation (14), our results also apply to PL functions.

## B.2 Valid and optimal preconditioners with singular matrices

In the main text, we defined valid preconditioners (Definition 4.1) only for positive definite matrices for ease of presentation. The notion of valid preconditioners can be extended to general positive semidefinite matrices. In the diagonal case, the convention $1/0 = +\infty$ is a useful mental model but can cause inconsistencies (such as $\infty \cdot 0$). To extend the notion of valid preconditioners to general positive semidefinite matrices, we can use the definition

**Definition B.1.** *A preconditioner $\mathbf{P} \succeq 0$ is* valid *if $\mathbf{P}^{1/2}\nabla^2 f(\mathbf{x})\mathbf{P}^{1/2} \preceq I$ for all $\mathbf{x} \in \mathbb{R}^d$.*

The above is well-defined for all positive semidefinite matrices. An alternative to arrive at a definition closer to Definition 4.1 is to consider the projection matrix $\Pi_{\mathbf{P}}$ onto the image of $\mathbf{P}$, given by $\Pi_{\mathbf{P}} = \mathbf{P}^{1/2}(\mathbf{P}^{1/2})^\dagger$ where $\mathbf{P}^\dagger$ is the Moore-Penrose pseudo-inverse of $\mathbf{P}$. Using that, one can show that $\mathbf{P}$ is *valid* (according to Definition B.1) if and only if

$$\Pi_{\mathbf{P}}\nabla^2 f(\mathbf{x})\Pi_{\mathbf{P}} \preceq \mathbf{P}^\dagger \qquad \text{for all } \mathbf{x} \in \mathbb{R}^d.$$

An example of a valid preconditioner that is covered by Definition B.1 but not 4.1 is the all-zeroes matrix. Definition B.1 can seamlessly replace 4.1, and all the results follow similarly. Moreover, notice that the optimization problem defining the optimal preconditioner (1) may not attain its minima on positive definite matrices when $f$ is not strongly convex. In this case, we can define an optimal

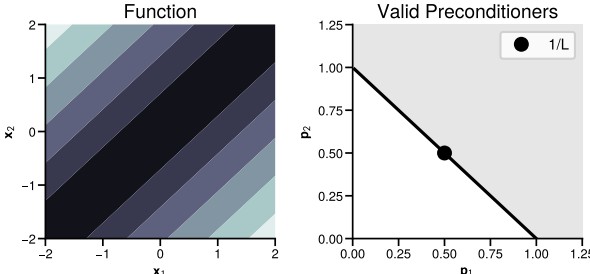

Figure 13: Set of valid diagonal preconditioners (step-sizes $\mathbf{p}_1$ and $\mathbf{p}_2$) for the quadratic in Equation (15). The set of valid preconditioners (Definition 4.1) is the white region in the right figure.

preconditioner as a limit point of a sequence that attains in the limit the value in (1) by replacing the minimum with an infimum. In this case, an optimal preconditioner may be singular, but the results in the main body also follow seamlessly using this definition. We decided to restrict our attention to non-singular preconditioners in the main paper for ease of exposition, since when $f$ is strongly-convex, an optimal preconditioner is always non-singular.

### B.3 Best competitive ratio achievable by the optimal preconditioner

In Section 3, we mentioned that the optimal preconditioner $\mathbf{P}_*$ could be only $1/d$-competitive. In fact, the competitive ratio of $\mathbf{P}_*$ can be arbitrarily bad. The reason for this is that the competitive ratio $\gamma$ does not compare against $\mathbf{P}_*$, but rather against any $\mathbf{P}$ in the set $\mathcal{S}$ of potentially valid preconditioners. Moreover, this definition only takes into account the norm $\|\nabla f(\mathbf{x})\|_{\mathbf{P}}$ at a *fixed* $\mathbf{x}$, while the optimal preconditioner needs to have large norm *for all* $\mathbf{x}$.

For example, consider the scalar step-size case. If our current interval of candidate step-sizes to try is $\mathcal{S} = [0, 1]$ but the optimal step-size $\alpha_*$ is small, let us say $\alpha_* = 1/10$, then $\alpha_*$ is only $1/10$-competitive in $\mathcal{S}$. The motivation for this definition of competitive ratio is that we cannot check whether $\alpha$ is large compared to $\alpha_*$ (as we do not know $\alpha_*$) but we can more easily ensure that a candidate step-size $\alpha$ is $\gamma$-competitive in $\mathcal{S}$ (for example $\alpha = 1/2$ is $1/2$-competitive in $[0, 1]$).

In the previous example, the bad competitive ratio of $\alpha_*$ in $\mathcal{S}$ was mostly due to the fact that $\mathcal{S}$ was large and that, for some $\mathbf{x}$, step sizes larger than $\alpha_*$ could satisfy the Armijo condition (3). Even if $\alpha_*$ is globally optimal, we could make more progress by using a larger step-size if they were to be accepted, and we have not yet ruled out those step-sizes. However, as $\mathcal{S}$ shrinks, it may eventually converge to the interval $[0, 1]$, in which case the optimal step-size $\alpha_*$ would be 1-competitive.

**In high dimensions however,** the optimal preconditioner can have a competitive ratio of $1/d$ even when comparing only against valid preconditioners.[5] This is because the competitive ratio is defined using the $\mathbf{P}$-norm of the gradient, and we need to take the direction of the gradient into account. For example, consider the quadratic function (illustrated in Figure 13)

$$f(\mathbf{x}) = \frac{1}{2}\langle \mathbf{x}, \mathbf{A}\mathbf{x} \rangle \qquad \text{where} \qquad \mathbf{A} = \begin{bmatrix} 1 & -1 \\ -1 & 1 \end{bmatrix}, \qquad (15)$$

with eigenvalues $\{2, 0\}$ as $\mathbf{A} = [-1, 1]^{\mathsf{T}}[-1, 1]$. The following three preconditioners are all valid:

$$\mathbf{P}_1 = \begin{bmatrix} 1 & 0 \\ 0 & 0 \end{bmatrix}, \qquad \mathbf{P}_2 = \begin{bmatrix} 0 & 0 \\ 0 & 1 \end{bmatrix}, \text{ and } \qquad \mathbf{P}_* = \begin{bmatrix} 1/2 & 0 \\ 0 & 1/2 \end{bmatrix}.$$

The preconditioner $\mathbf{P}_1$ takes the largest possible step-size in the first coordinate and ignores the second, while $\mathbf{P}_2$ does the opposite. They are not good global preconditioners, as each ignores one coordinate. Yet, they can make much more progress (i.e., the objective value may decrease more) than the optimal preconditioner $\mathbf{P}_*$ if the gradient is very skewed towards one coordinate. This implies that

---

[5]How small the set $\mathcal{S}_t$ can get is bounded by construction. The cutting plane procedure in Sections 4 and 5 only remove invalid preconditioners. The valid preconditioners contained in the initial set $\mathcal{S}_0$ will always be in $\mathcal{S}_t$, along with possibly more preconditioners that have not been deemed invalid over the course of optimization.

$\mathbf{P}_*$ may be only $1/2$-competitive in $\{\mathbf{P}_1, \mathbf{P}_2\}$ for some $\mathbf{x}$ since

$$\text{if } \nabla f(\mathbf{x}) = \begin{bmatrix} 1 \\ 0 \end{bmatrix}, \text{ then } \quad \|\nabla f(\mathbf{x})\|_{\mathbf{P}_1}^2 = 1, \quad \|\nabla f(\mathbf{x})\|_{\mathbf{P}_2}^2 = 0, \quad \|\nabla f(\mathbf{x})\|_{\mathbf{P}_*}^2 = 1/2,$$

$$\text{and } \quad \text{if } \nabla f(\mathbf{x}) = \begin{bmatrix} 0 \\ 1 \end{bmatrix}, \text{ then } \quad \|\nabla f(\mathbf{x})\|_{\mathbf{P}_1}^2 = 0, \quad \|\nabla f(\mathbf{x})\|_{\mathbf{P}_2}^2 = 1, \quad \|\nabla f(\mathbf{x})\|_{\mathbf{P}_*}^2 = 1/2.$$

The preconditioner $\mathbf{P}_*$ is still a better choice globally (i.e, for all $\mathbf{x}$) since it ensures optimal worst-case linear rate in preconditioned gradient descent. But there are better preconditioners that depend on the current gradient. We exploit this in the ellipsoid variant of multidimensional backtracking to improve our competitive ratio. We backtrack from the preconditioner that maximizes the local progress guarantees to ensure a $1/\sqrt{d}$ competitive ratio, while ensuring volume shrinkage of the set of candidate preconditioners when we call CUT, if the preconditioner fails the Armijo condition.

## C Separating hyperplanes

In this section, we prove Propositions 4.2 and 4.3 on existence and strengthening of separating hyperplanes for valid preconditioners.

**General idea.** Let us start with a summary of the separating hyperplanes used to search for good preconditioners as discussed in Sections 3 and 4. The goal of the separating hyperplanes is to give us ways to shrink the initial set of potential preconditioners $\mathcal{S}$ to narrow in on valid preconditioners using the cutting-plane methods in Section 5. At each iteration we are looking for preconditioners $\mathbf{P}$ that satisfy the Armijo condition at $\mathbf{x}$ given by

$$f(\mathbf{x} - \mathbf{P}\nabla f(\mathbf{x})) \leq f(\mathbf{x}) - \frac{1}{2}\|\nabla f(\mathbf{x})\|_{\mathbf{P}}^2.$$

If $\mathbf{P}$ fails the Armijo condition, we conclude that $\mathbf{P}$ is invalid. To obtain more information, we look at the condition as a function of the (diagonal of the) preconditioner, and define the gap function at $\mathbf{x}$,

$$h(\mathbf{p}) := f(\mathbf{x} - \mathrm{Diag}(\mathbf{p})\nabla f(\mathbf{x})) - f(\mathbf{x}) + \frac{1}{2}\|\nabla f(\mathbf{x})\|_{\mathrm{Diag}(\mathbf{p})}^2, \qquad \forall \mathbf{p} \in \mathbb{R}_{\geq 0}^d.$$

Then, $h(\mathbf{p}) \leq 0$ if $\mathbf{P} = \mathrm{Diag}(\mathbf{p})$ satisfies the Armijo condition at $\mathbf{x}$, and $h(\mathbf{p}) > 0$ otherwise. Any preconditioner $\mathrm{Diag}(\mathbf{q})$ such that $h(\mathbf{q}) > 0$ is guaranteed to be invalid. We can use the gradient of $h$ at $\mathbf{p}$ and convexity to find a half-space such that one side contains only preconditioners with $h(\mathbf{p}) > 0$. In this section, we show how to construct such half-space, and strengthen them using the partial order on matrices, which is needed to ensure volume shrinkage of our cutting plane methods.

### C.1 Stronger hyperplanes

In the main body we presented the strengthening of separating hyperplanes via truncation (Proposition 4.3) after the result of existence of separating hyperplanes (Proposition 4.2). Here, we prove a more general lemma on strengthening half-spaces of invalid preconditioners first, as it is useful in simplifying the proof of Proposition 4.2. Proposition 4.3 follows directly from the following lemma.

**Lemma C.1.** *Let $\mathcal{H}_{\mathbf{v},\alpha}$ be the intersection of the non-negative orthant $\mathbb{R}_{\geq 0}^d$ and the half-space defined by the vector $\mathbf{v} \in \mathbb{R}^d$ and coefficient $\alpha > 0$,*

$$\mathcal{H}_{\mathbf{v},\alpha} := \{\, \mathbf{p} \in \mathbb{R}_{\geq 0}^d : \langle \mathbf{v}, \mathbf{p} \rangle > \alpha \}.$$

*Define $\mathbf{u} := \max\{\mathbf{v}, 0\}$ and let $\mathcal{H}_{\mathbf{u},\alpha}$ be defined similarly as above, that is,*

$$\mathcal{H}_{\mathbf{u},\alpha} := \{\, \mathbf{p} \in \mathbb{R}_{\geq 0}^d : \langle \mathbf{u}, \mathbf{p} \rangle > \alpha \}.$$

*If $\mathcal{H}_{\mathbf{v},\alpha}$ only contains diagonals of invalid preconditioners, that is, $\mathrm{Diag}(\mathbf{p})$ is invalid for any $\mathbf{p} \in \mathcal{H}_{\mathbf{v}}$, Then $\mathcal{H}_{\mathbf{v},\alpha} \subseteq \mathcal{H}_{\mathbf{u},\alpha}$ and $\mathcal{H}_{\mathbf{u},\alpha}$ only contains diagonals of invalid preconditioners.*

*Proof.* **Inclusion $\mathcal{H}_{\mathbf{v},\alpha} \subseteq \mathcal{H}_{\mathbf{u},\alpha}$.** We have that $\langle \mathbf{p}, \mathbf{v} \rangle > \alpha$ implies $\langle \mathbf{p}, \mathbf{u} \rangle > \alpha$ for any $\mathbf{p} \in \mathbb{R}_{\geq 0}^d$ since

$$\langle \mathbf{v}, \mathbf{p} \rangle = \sum_{i\,:\,\mathbf{v}[i] \geq 0} \mathbf{v}[i]\mathbf{p}[i] + \sum_{i\,:\,\mathbf{v}[i] < 0} \mathbf{v}[i]\mathbf{p}[i] \leq \sum_{i\,:\,\mathbf{v}[i] \geq 0} \mathbf{v}[i]\mathbf{p}[i] = \sum_{i\,:\,\mathbf{v}[i] \geq 0} \mathbf{u}[i]\mathbf{p}[i] = \langle \mathbf{u}, \mathbf{p} \rangle.$$

**$\mathcal{H}_{\mathbf{u},\alpha}$ only contains invalid diagonals.** Let $\mathbf{p_u} \in \mathcal{H}_{\mathbf{u},\alpha}$. We can show that $\mathrm{Diag}(\mathbf{p_u})$ is invalid by finding $\mathbf{p_v} \in \mathcal{H}_{\mathbf{v},\alpha}$ such that $\mathrm{Diag}(\mathbf{p_v}) \preceq \mathrm{Diag}(\mathbf{p_u})$. Since $\mathrm{Diag}(\mathbf{p_v})$ is invalid by assumption, this would imply that $\mathrm{Diag}(\mathbf{p_u})$ is also invalid. To find $\mathbf{p_v}$, we can truncate the entries of $\mathbf{p_u}$ as

$$\mathbf{p_v}[i] := \begin{cases} \mathbf{p_u}[i] & \text{if } \mathbf{v}[i] \geq 0 \\ 0 & \text{otherwise}, \end{cases} \qquad \forall i \in \{1, \ldots, d\}.$$

Then $\mathbf{p_v} \in \mathcal{H}_{\mathbf{v},\alpha}$ since $\alpha < \langle \mathbf{u}, \mathbf{p_u} \rangle = \langle \mathbf{u}, \mathbf{p_v} \rangle = \langle \mathbf{v}, \mathbf{p_v} \rangle.$ [6] and $\mathrm{Diag}(\mathbf{p_u}) \succeq \mathrm{Diag}(\mathbf{p_v})$, as desired. $\qquad\square$

---

[6]One may worry that our original definition of valid preconditioners has a division by $0$ if any entry of the preconditioner is $0$ as a preconditioner is valid if $\nabla^2 f(\mathbf{x}) \preceq \mathbf{P}^{-1}$ (Definition 4.1). It is enough to use the convention that $1/0 = +\infty$, although this might lead to inconsistencies. In Appendix B.2 we discuss a more general definition without the use of infinities.

## C.2 Separating hyperplanes for invalid preconditioners

We are now in position to prove Proposition 4.2.

*Proof of Proposition 4.2.* Throughout the proof, we shall denote by $\mathbf{P}$ the matrix $\mathrm{Diag}(\mathbf{p})$. If $f$ is convex, then $h$ also is since the map $\mathbf{p} \in \mathbb{R}^d_{\geq 0} \mapsto f(\mathbf{x} - \mathbf{P}\nabla f(\mathbf{x}))$ is the composition of an affine transformation and a convex function, and $\|\nabla f(\mathbf{x})\|^2_{\mathbf{P}} = \langle \nabla f(\mathbf{x}), \mathrm{Diag}(\mathbf{p})\nabla f(\mathbf{x})\rangle$ is linear in $\mathbf{p}$. Convexity of $h$ yields the inequality

$$h(\mathbf{p}) \geq h(\mathbf{q}) + \langle \nabla h(\mathbf{q}), \mathbf{p} - \mathbf{q}\rangle, \qquad \forall \mathbf{p} \in \mathbb{R}^d_{\geq 0}.$$

This implies that if $\mathbf{p}$ is such that $h(\mathbf{q}) + \langle \nabla h(\mathbf{q}), \mathbf{p} - \mathbf{q}\rangle > 0$, then $h(\mathbf{p}) > 0$, which implies that $\mathrm{Diag}(\mathbf{p})$ is an invalid preconditioner. Rearranging we conclude that $\mathrm{Diag}(\mathbf{p})$ is invalid for all $\mathbf{p}$ in the set in (5), i.e., in

$$\{\, \mathbf{p} \in \mathbb{R}^d_{\geq 0} : \langle \nabla h(\mathbf{q}), \mathbf{p}\rangle > \langle \nabla h(\mathbf{q}), \mathbf{q}\rangle - h(\mathbf{q})\} \tag{16}$$

We express the above half-space as

$$\mathcal{H}_>(\mathbf{v}) = \{\mathbf{p} : \langle \mathbf{p}, \mathbf{v}\rangle > 1\} \text{ for } \mathbf{v} := \frac{\nabla h(\mathbf{q})}{(\langle \nabla h(\mathbf{q}), \mathbf{q}\rangle - h(\mathbf{q}))}.$$

Yet, for $\mathcal{H}_>(\mathbf{v})$ to be equivalent to the set in (16) or even to be well-defined, we need to ensure $\langle \nabla h(\mathbf{q}), \mathbf{q}\rangle - h(\mathbf{q}) > 0$. To see that this holds, note first that by convexity of $h$ and that fact that $h(0) = 0$ we have

$$h(0) \geq h(\mathbf{q}) + \langle \nabla h(\mathbf{q}), 0 - \mathbf{q}\rangle \implies \langle \nabla h(\mathbf{q}), \mathbf{q} - 0\rangle - h(\mathbf{q}) \geq -h(0) = 0$$

To show that the last inequality is strict, assume that $\langle \nabla h(\mathbf{q}), \mathbf{q} - 0\rangle - h(\mathbf{q}) = 0$ for the sake of contradiction. By Lemma C.1, the half-space $\mathcal{H} := \{\, \mathbf{p} \in \mathbb{R}^d_{\geq 0} : \langle [\nabla h(\mathbf{x})]_+, \mathbf{p}\rangle > 0\}$ contains only diagonals of invalid preconditioners, where $[\nabla h(\mathbf{x})]_+ := \max\{\nabla h(\mathbf{x}), 0\}$ entry wise. However, $(1/L)\mathbf{1} \in \mathcal{H}$ as $[\nabla h(\mathbf{x})]_+ \geq 0$ and should be invalid, which is a contradiction since $f$ is $L$-smooth and $1/L\mathbf{I}$ is valid. Therefore, $\langle \nabla h(\mathbf{q}), \mathbf{q} - 0\rangle - h(\mathbf{q}) > 0$.

Finally, we can write $\mathbf{v}$ in terms of $f$ and $\mathbf{Q}$. To do so, first define $\mathbf{x}^+ := \mathbf{x} - \mathbf{Q}\nabla f(\mathbf{x})$, and the gradients of $f$ at different points by $\mathbf{g} := \nabla f(\mathbf{x})$ and $\mathbf{g}^+ := \nabla f(\mathbf{x}^+)$. Then, by the chain-rule,

$$\nabla h(\mathbf{q}) = -\nabla f(\mathbf{x} - \mathbf{Q}\nabla f(\mathbf{x})) \odot \nabla f(\mathbf{x}) + \frac{1}{2}\nabla f(\mathbf{x}) \odot \nabla f(\mathbf{x}) = -\mathbf{g}^+ \odot \mathbf{g} + \frac{1}{2}\mathbf{g} \odot \mathbf{g},$$

which implies

$$\langle \nabla h(\mathbf{q}), \mathbf{q}\rangle - h(\mathbf{q}) = -\langle \mathbf{g}^+, \mathbf{Qg}\rangle + \frac{1}{2}\langle \mathbf{g}, \mathbf{Qg}\rangle - f(\mathbf{x}^+) + f(\mathbf{x}) + \frac{1}{2}\langle \mathbf{g}, \mathbf{Qg}\rangle$$
$$= f(\mathbf{x}) - \langle \mathbf{g}^+, \mathbf{Qg}\rangle - f(\mathbf{x}^+).$$

Plugging these equations in the definition of $\mathbf{v}$ yields

$$\mathbf{v} = \frac{\nabla h(\mathbf{q})}{\langle \nabla h(\mathbf{q}), \mathbf{q}\rangle - h(\mathbf{q})} = \frac{(\frac{1}{2}\mathbf{g} - \mathbf{g}^+) \odot \mathbf{g}}{f(\mathbf{x}) - \langle \mathbf{g}^+, \mathbf{Qg}\rangle - f(\mathbf{x}^+)}. \qquad \square$$

**Remark on assumptions of Proposition 4.2.** One may have noticed that we never use the assumption that $\mathbf{Q}$ fails the Armijo condition (i.e., that $h(\mathbf{q}) > 0$) in the proof of the proposition. In fact, the proposition holds for any $\mathbf{q} \in \mathbb{R}^d_{\geq 0}$. However, and crucially for our application, we have that $\mathbf{q}$ is in the half-space $\mathcal{H}_>(\mathbf{u})$ of invalid diagonals from Proposition 4.2. In multidimensional backtracking, $\mathbf{q}$ is the diagonal of a preconditioner $\mathrm{Diag}(\mathbf{q})$ that failed the Armijo condition $h(\mathbf{q}) > 0$. Since $\mathbf{q}$ is close to the origin in multidimensional backtracking, we can ensure the half-space $\mathcal{H}_>(\mathbf{u})$ contains a significant portion of our current set of candidate preconditioners, leading to significant shrinkage of the set of candidate preconditioners whenever CUT is invoked.

# D Cutting-plane methods

## D.1 Boxes

Given a box $\mathcal{B}(\mathbf{b})$ for some $\mathbf{b} \in \mathbb{R}^d_{\geq 0}$ and a vector $\mathbf{u} \in \mathbb{R}^d_{\geq 0}$, our cutting plane method needs to find a box $\mathcal{B}(\mathbf{b}^+)$ that contains $\mathcal{B}(\mathbf{b}) \cap \mathcal{H}_>(\mathbf{u})$ which, hopefully, has smaller volume than $\mathcal{B}(\mathbf{b})$.

The next lemma gives a formula for the *minimum volume* box for any $\mathbf{u}$, which is used in the main text to define CUT in Equation (7). Moreover, we show that if the half-space $\mathcal{H}_>(\mathbf{u})$ is close enough to the origin (since otherwise we might have $\mathbf{b}^+ = \mathbf{b}$), then we have a significant volume decrease.

**Lemma D.1.** *Let $\mathbf{b} \in \mathbb{R}^d_{\geq 0}$ and $\mathbf{q} \in \mathcal{B}(\mathbf{b})$. Let $\mathbf{u} \in \mathbb{R}^d_{\geq 0}$. Then the box $\mathcal{B}(\mathbf{b}^+)$ with minimum volume that contains $\mathcal{B}(\mathbf{b}) \cap \mathcal{H}_\leq(\mathbf{u})$ is given by (using the convention that $1/\mathbf{u}[i] = +\infty$ if $\mathbf{u}[i] = 0$)*

$$\mathbf{b}^+[i] := \min\{\mathbf{b}[i], 1/\mathbf{u}[i]\}, \qquad \forall i \in \{1, \ldots, d\}, \tag{17}$$

*Moreover, if $(1/2d) \cdot \mathbf{b}$ is excluded by the half-space, that is, $\mathbf{b} \in \mathcal{H}_>(\mathbf{u})$, then $\mathrm{Vol}(\mathcal{B}(\mathbf{b}^+)) \leq (1/(d+1)) \, \mathrm{Vol}(\mathcal{B}(\mathbf{b}^+))$.*

*Proof.* **Formula for $\mathbf{b}^+$.** Finding the minimum volume box containing $\mathcal{B}(\mathbf{b}) \cap \mathcal{H}_\leq(\mathbf{u})$,

$$\mathbf{b}^+ = \arg\min_{\mathbf{c} \in \mathbb{R}^d} \mathrm{Vol}(\mathcal{B}(\mathbf{c})) \quad \text{s.t. } \mathcal{B}(\mathbf{b}) \cap \mathcal{H}_\leq(\mathbf{u}) \subseteq \mathcal{B}(\mathbf{c}),$$

is equivalent to finding the solution to the following optimization problem:

$$\mathbf{b}^+ = \arg\min_{\mathbf{c} \in \mathbb{R}^d} \prod_i \mathbf{c}[i] \quad \text{s.t. } \max_{\mathbf{p} \in \mathcal{B}(\mathbf{c}) \cap \mathcal{H}_\leq(\mathbf{u})} \mathbf{p}[i] \leq \mathbf{c}[i] \text{ for each } i \in \{1, \ldots, d\}.$$

As the constraints separate over the coordinates, the minimization can be done for each coordinate separately. As the function is increasing in $\mathbf{c}[i]$, the minimum is achieved by making all the constraints tight, which giver the formula for $\mathbf{b}^+$ in the statement of the lemma.

**Volume decrease.** Let us prove the second part of the statement. Thus, assume for the remainder of the proof that $(1/2d) \cdot \mathbf{b} \in \mathcal{H}_>(\mathbf{u})$. We first show that $\mathrm{Vol}(\mathcal{B}(\mathbf{b}^+)) \leq (1/(d+1)) \, \mathrm{Vol}(\mathcal{B}(\mathbf{b}^+))$ if we assume that the update from $\mathcal{B}(\mathbf{b})$ to $\mathcal{B}(\mathbf{b}^+)$ shrinks the box in only one coordinate, i.e.,

$$\mathcal{I} := \{\, i \in [d] : \mathbf{b}[i] > 1/\mathbf{u}[i] \,\} = \{\, i \in [d] : \mathbf{b}^+[i] \neq \mathbf{b}[i] \,\} \text{ has exactly one element.} \tag{18}$$

Assume the above holds and $\mathcal{I} = \{j\}$. Then, as $(1/2d) \cdot \mathbf{b} \in \mathcal{H}_>(\mathbf{u})$ implies $\langle \mathbf{u}, (1/2d)\mathbf{b} \rangle > 1$,

$$1 < \langle \mathbf{u}, (1/2d)\mathbf{b} \rangle \leq \frac{1}{2d}(\mathbf{u}[j]\mathbf{b}[j] + d - 1) \implies (d+1)\frac{1}{\mathbf{u}[j]} \leq \mathbf{b}[j].$$

This together with the fact that $\mathbf{b}^+[i] = \mathbf{b}[i]$ for all $i \neq j$ and $\mathbf{b}^+[j] = 1/\mathbf{u}[j]$ yields

$$\mathrm{Vol}(\mathcal{B}(\mathbf{b}^+)) = \prod_{i=1}^d \mathbf{b}^+[i] = \frac{1}{\mathbf{u}[j]} \cdot \prod_{i \neq j} \mathbf{b}[i] \leq \frac{1}{d+1} \prod_{i=1}^d \mathbf{b}[i] = \frac{1}{d+1} \, \mathrm{Vol}(\mathcal{B}(\mathbf{b})).$$

To complete the proof, we only need to show we may assume (18) holds. Assume the opposite, that is, that there are two distinct coordinates that shrink from $\mathbf{b}^+$ to $\mathbf{b}$. We will show that the volume shrinks more, meaning the above bound also applies. Formally, assume there are $j, k \in \mathcal{I}$ that are distinct. For this part, it will be useful to denote by $\mathbf{b}^+(\mathbf{u})$ the point defined in Equation (17) for a given vector $\mathbf{u}$. We will show we can construct $\mathbf{u}' \in \mathbb{R}^d_{\geq 0}$ such that $\mathrm{Vol}(\mathbf{b}^+(\mathbf{u})) \leq \mathrm{Vol}(\mathbf{b}^+(\mathbf{u}'))$ while maintaining the property $(1/2d)\mathbf{b} \in \mathcal{H}_>(\mathbf{u}')$ and such that $\mathbf{b}^+(\mathbf{u}')[i] \neq \mathbf{b}[i]$ for all $i \in \mathcal{I} \setminus \{j\}$, which makes (18) follow by induction. Indeed, define $\mathbf{u}' \in \mathbb{R}^d_{\geq 0}$ by

$$\mathbf{u}'[i] := \mathbf{u}[i] \text{ for } i \notin \{j, k\}, \quad \mathbf{u}'[j] := \frac{1}{\mathbf{b}[j]}, \quad \text{and } \mathbf{u}'[k] := \mathbf{u}[k] + \frac{\mathbf{b}[j]}{\mathbf{b}[k]}\left(\mathbf{u}[j] - \frac{1}{\mathbf{b}[j]}\right). \tag{19}$$

First, note that $(1/2d)\mathbf{b} \in \mathcal{H}_>(\mathbf{u}')$ since

$$\langle \mathbf{u}' - \mathbf{u}, \mathbf{b} \rangle = \mathbf{b}[j](\mathbf{u}'[j] - \mathbf{u}[j]) + (\mathbf{u}'[k] - \mathbf{u}[k])\mathbf{b}[k]$$

$$= \mathbf{b}[j]\left(\frac{1}{\mathbf{b}[j]} - \mathbf{u}[j]\right) + \left(\frac{\mathbf{b}[j]}{\mathbf{b}[k]}\left(\mathbf{u}[j] - \frac{1}{\mathbf{b}[j]}\right)\right)\mathbf{b}[k] = 0$$

and, thus, $1 < \langle \mathbf{u}, (1/2d)\mathbf{b} \rangle = \langle \mathbf{u}', (1/2d)\mathbf{b} \rangle$. Let us now show that $\mathrm{Vol}(\mathcal{B}(\mathbf{b}^+(\mathbf{u}))) \leq \mathrm{Vol}(\mathcal{B}(\mathbf{b}^+(\mathbf{u}')))$. Since $\mathbf{b}^+(\mathbf{u})[i] = \mathbf{b}^+(\mathbf{u}')[i]$ for $i \notin \{j, k\}$, we have

$$
\begin{aligned}
\frac{\mathrm{Vol}(\mathcal{B}(\mathbf{b}^+(\mathbf{u})))}{\mathrm{Vol}(\mathcal{B}(\mathbf{b}^+(\mathbf{u}')))} &= \frac{\mathbf{b}^+(\mathbf{u})[j]}{\mathbf{b}^+(\mathbf{u}')[j]} \cdot \frac{\mathbf{b}^+(\mathbf{u})[k]}{\mathbf{b}^+(\mathbf{u}')[k]} \\
&= \frac{\min(\mathbf{b}[j], 1/\mathbf{u}[j])}{\min(\mathbf{b}[j], 1/\mathbf{u}'[j])} \cdot \frac{\min(\mathbf{b}[k], 1/\mathbf{u}[k])}{\min(\mathbf{b}[k], 1/\mathbf{u}'[k])} \\
&= \frac{1/\mathbf{u}[j]}{\mathbf{b}[j]} \cdot \frac{1/\mathbf{u}[k]}{1/\mathbf{u}'[k]} \qquad\qquad \text{(since } j, k \in \mathcal{I} \text{ and by (19))} \\
&= \frac{1}{\mathbf{b}[j]\mathbf{u}[j]} \cdot \frac{1}{\mathbf{u}[k]}\left(\mathbf{u}[k] + \frac{\mathbf{b}[j]}{\mathbf{b}[k]}\left(\mathbf{u}[j] - \frac{1}{\mathbf{b}[j]}\right)\right) \\
&= \frac{1}{\mathbf{b}[j]\mathbf{u}[j]} \cdot \frac{1}{\mathbf{u}[k]\mathbf{b}[k]}(\mathbf{b}[k]\mathbf{u}[k] + \mathbf{b}[j]\mathbf{u}[j] - 1).
\end{aligned}
$$

To get that $\mathrm{Vol}(\mathbf{b}^+(\mathbf{u})) \leq \mathrm{Vol}(\mathbf{b}^+(\mathbf{u}'))$, we can show that last line is bounded by $< 1$. Using the substitution $\alpha := \mathbf{b}[j]\mathbf{u}[j]$ and $\beta := \mathbf{b}[k]\mathbf{u}[k]$, we want to show that

$$
\frac{\alpha + \beta - 1}{\alpha\beta} < 1 \iff \alpha\beta - \alpha - \beta + 1 > 0 \iff (\alpha - 1)(\beta - 1) > 0.
$$

This holds if $\alpha > 1$ and $\beta > 1$, is implied by $j, k \in \mathcal{I}$ since $\alpha = \mathbf{b}[j]\mathbf{u}[j] > 1$ and $\beta = \mathbf{b}[k]\mathbf{u}[k] > 1$. A simple induction shows we may assume (18) holds. To see that $(\alpha + \beta - 1)/\alpha\beta < 1$, note that $\quad\square$

Equipped with the above lemma, we are in position to prove Theorem 5.1.

*Proof of Theorem 5.1.* Property (a), holds by induction because, for any $\mathbf{u}_t$ used in a call to CUT, we have $\mathbf{P}^* \in \mathcal{H}_\leq(\mathbf{u}_t)$ since $\mathbf{P}^*$ is valid and since by Proposition 4.2 the half-space $\mathcal{H}_\leq(\mathbf{u}_t)$ contains only diagonals of invalid preconditioners. For (b), fix $t \in \{1, \ldots, T\}$ and recall that in this case we have $\mathcal{S}_t = \{\mathrm{Diag}(\mathbf{p}) : \mathbf{p} \in \mathcal{B}(\mathbf{b}_t)\}$ and $\mathbf{P}_t = (1/2d) \cdot \mathrm{Diag}(\mathbf{b}_t)$. The competitive ratio of $1/2d$ follows since $\mathrm{Diag}(\mathbf{b}_t)$ is the preconditioner that maximizes $\|\nabla f(\mathbf{x}_t)\|_\mathbf{P}$ for $\mathbf{P} \in \mathcal{B}(\mathbf{b}_t)$. Finally, for (c) by Lemma D.1 we have that every call to CUT makes the volume of the set decrease by $1/c := 1/(d+1)$. Moreover, one can easily verify that $\mathbf{b}_t[i] \geq \min\{1/L, \mathbf{b}_0[i]\}$ for all $i \in \{1, \ldots, d\}$ since $\mathcal{B}((1/L)\mathbb{1})$ contains only diagonals of valid preconditioners. Therefore, for $\mathbf{b}_{\min}[i] := \min\{1/L, \mathbf{b}_0[i]\}$, the volume of $\mathcal{B}(\mathbf{b}_t)$ cannot be smaller than $\mathcal{B}(\mathbf{b}_{\min})$ for all iteration $t$. Therefore, the number of times CUT is invoked is no more than

$$
\log_c\left(\frac{\mathrm{Vol}(\mathcal{B}(\mathbf{b}_0))}{\mathrm{Vol}(\mathcal{B}(\mathbf{b}_{\min}))}\right) = \log_c\left(\prod_{i=1}^d \frac{\mathbf{b}_0[i]}{\mathbf{b}_{\min}[i]}\right) \leq \log_c((\|\mathbf{b}_0\|_\infty L)^d) = d\log_c(\|\mathbf{b}_0\|_\infty L).
$$

as desired. $\quad\square$

## D.2 Axis-aligned ellipsoids

We now analyze the cutting-plane method using axis-aligned ellipsoids. Interestingly, the results that we prove in this sections are connected to some of the results from Goemans et al. (2009) via polarity theory. We defer a discussion on this connection to the end of this section.

Different from the main body, it will be helpful for the analysis of the method and proofs of the results to not restrict ellipsoids to the non-negative orthant, as was done in the main text for ease of exposition. For any symmetric positive definite matrix $\mathbf{A} \in \mathbb{R}^{d \times d}$, define the *ellipsoid* given by $\mathbf{A}$ by

$$
\mathcal{E}(\mathbf{A}) := \{\mathbf{x} \in \mathbb{R}^d : \langle \mathbf{x}, \mathbf{A}\mathbf{x} \rangle \leq 1\}.
$$

When $\mathbf{A}$ is diagonal, we say that $\mathcal{E}(\mathbf{A})$ is *axis-aligned*. Moreover, we may slightly overload our notation by defining $\mathcal{E}(\mathbf{a}) := \mathcal{E}(\mathrm{Diag}(\mathbf{a}))$.

**General ellipsoids.** Although we are ultimately interested in working solely with ellipsoids defined by diagonal matrices, we will start by looking at more general ellipsoids, and then exploit symmetry in our case to derive the result in Lemma 5.2. We start with an ellipsoid $\mathcal{E}(\mathbf{A})$ where $\mathbf{A}$ is a positive definite matrix. Then, given a vector $\mathbf{u} \in \mathbb{R}^d$, we are interested in finding an ellipsoid the intersection

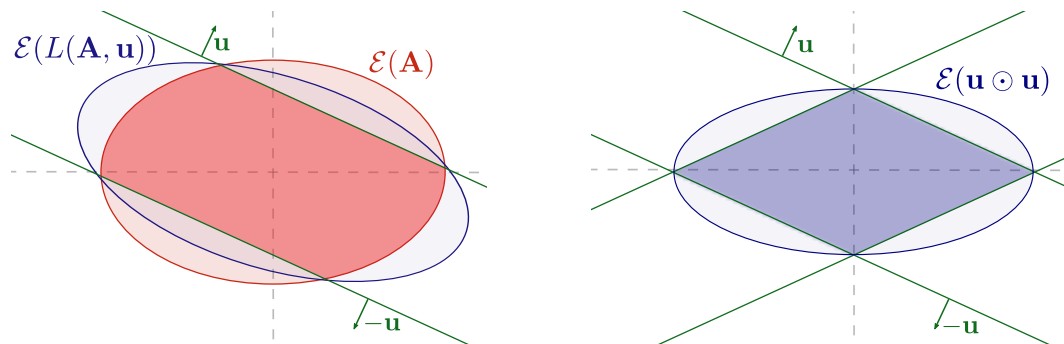

Figure 14: Illustration of the ellipsoids in Theorem D.2 in the left. In the right an illustration of the symmetrized intersection of halfspaces $\mathcal{S}(\mathbf{u})$ used in the proof of Lemma 5.2 together with the ellipsoid $\mathcal{E}(\mathbf{u} \odot \mathbf{u})$ used in the convex combination in the lemma.

of $\mathcal{E}(\mathbf{A})$ with the half-spaces defined by $\mathbf{u}$ and $-\mathbf{u}$ that contain the origin, that is, the set

$$\mathcal{E}(\mathbf{A}) \cap \{\, \mathbf{x} \in \mathbb{R}^d : \langle \mathbf{x}, \mathbf{u} \rangle < 1 \,\} \cap \{\, \mathbf{x} \in \mathbb{R}^d : -\langle \mathbf{x}, \mathbf{u} \rangle < 1 \,\} = \mathcal{E}(\mathbf{A}) \cap \{\, \mathbf{x} \in \mathbb{R}^d : |\langle \mathbf{x}, \mathbf{u} \rangle| < 1 \,\}.$$

The following theorem shows how to find an ellipsoid that contains the above intersection, and how to guarantee its volume is smaller than $\mathcal{E}(\mathbf{A})$ if $\mathbf{u}$ is large enough. Interestingly, note that

$$\{\, \mathbf{x} \in \mathbb{R}^d : |\langle \mathbf{x}, \mathbf{u} \rangle| < 1 \,\} = \{\, \mathbf{x} \in \mathbb{R}^d : (\langle \mathbf{x}, \mathbf{u} \rangle)^2 < 1 \,\} = \mathcal{E}(\mathbf{u}\mathbf{u}^{\mathsf{T}}).$$

The set $\mathcal{E}(\mathbf{u}\mathbf{u}^{\mathsf{T}})$ is a degenerate ellipsoid, in the sense that it is not a compact set, and any $\mathbf{p}$ orthogonal to $\mathbf{u}$ is contained in $\mathcal{E}(\mathbf{u}\mathbf{u}^{\top})$. Still, the next theorem shows how to find a convex combination of $\mathcal{E}(\mathbf{A})$ and $\mathcal{E}(\mathbf{u}\mathbf{u}^{\mathsf{T}})$—which always contains $\mathcal{E}(\mathbf{A}) \cap \mathcal{E}(\mathbf{u}\mathbf{u}^{\mathsf{T}})$—that is guaranteed to have volume smaller than $\mathcal{E}(\mathbf{A})$ if $\mathbf{u}$ is large enough. The following result can be seen as the polar result of Goemans et al. (2009, Lemma 2).

**Theorem D.2.** *Let* $\mathbf{A} \in \mathbb{R}^{d \times d}$ *be positive definite and let* $\mathbf{u} \in \mathbb{R}^d$. *Let* $\lambda \in (0, 1)$ *and define*

$$L(\mathbf{A}, \mathbf{u}) := \lambda \mathbf{A} + (1 - \lambda)\mathbf{u}\mathbf{u}^{\mathsf{T}}.$$

*Then* $\mathcal{E}(\mathbf{A}) \cap \mathcal{E}(\mathbf{u}\mathbf{u}^{\mathsf{T}}) \subseteq \mathcal{E}(L(\mathbf{A}, \mathbf{u}))$ *and*

$$\mathrm{Vol}(\mathcal{E}(L(\mathbf{A}, \mathbf{u}))) = \sqrt{\frac{\lambda}{\lambda + (1 - \lambda) \cdot \ell} \cdot \frac{1}{\lambda^d}} \cdot \mathrm{Vol}(\mathcal{E}(\mathbf{A}))$$

*In particular, if* $\ell := \|\mathbf{u}\|_{\mathbf{A}^{-1}}^2 > d$ *and*

$$\lambda = \frac{\ell}{d} \cdot \frac{d-1}{\ell - 1}, \tag{20}$$

*then* $\lambda \in (0, 1)$ *and* $\mathrm{Vol}(\mathcal{E}(L(\mathbf{A}, \mathbf{u}))) = \nu_d(\mathbf{u}) \mathrm{Vol}(\mathcal{E}(\mathbf{A}))$ *where*

$$\nu_d(\mathbf{u}) = \sqrt{\frac{1}{\lambda^d} \cdot \frac{d-1}{\ell - 1}} = \left(\frac{d}{\ell}\right)^{d/2} \left(\frac{\ell - 1}{d - 1}\right)^{(d-1)/2} \in (0, 1). \tag{21}$$

*Proof.* First, note that for any $\mathbf{p} \in \mathcal{E}(\mathbf{A}) \cap \mathcal{E}(\mathbf{u}\mathbf{u}^{\mathsf{T}})$ and any $\lambda \in (0, 1)$ we have

$$\langle \mathbf{p}, L(\mathbf{A}, \mathbf{u})\mathbf{p} \rangle = \lambda \langle \mathbf{p}, \mathbf{A}\mathbf{p} \rangle + (1 - \lambda)\langle \mathbf{p}, \mathbf{u} \rangle \le \lambda + (1 - \lambda) = 1.$$

Thus, $\mathcal{E}(L(\mathbf{A}, \mathbf{u})) \subseteq \mathcal{E}(\mathbf{A}) \cap \mathcal{E}(\mathbf{u}\mathbf{u}^{\mathsf{T}})$. For the volume decrease, recall that for ellipsoids $\mathcal{E}(\mathbf{A})$ we have $\mathrm{Vol}(\mathcal{E}(\mathbf{A})) = V_d/\sqrt{\det(\mathbf{A})}$ where $V_d$ is the volume of the unit sphere in $\mathbb{R}^d$. By the matrix-determinant lemma, we have

$$\det(L(\mathbf{A}, \mathbf{u})) = \left(1 + \frac{1 - \lambda}{\lambda} \cdot \langle \mathbf{u}, \mathbf{A}^{-1}\mathbf{u} \rangle\right)\det(\lambda \mathbf{A}) = \left(1 + \frac{1 - \lambda}{\lambda} \cdot \ell\right)\lambda^d \det(\mathbf{A}).$$

Therefore,

$$\text{Vol}(\mathcal{E}(L(\mathbf{A}, \mathbf{u}))) = \sqrt{\frac{1}{\left(1 + \frac{1-\lambda}{\lambda} \cdot \ell\right)} \cdot \frac{1}{\lambda^d}} \cdot \text{Vol}(\mathcal{E}(\mathbf{A})) = \sqrt{\frac{\lambda}{\lambda + (1-\lambda) \cdot \ell} \cdot \frac{1}{\lambda^d}} \cdot \text{Vol}(\mathcal{E}(\mathbf{A})).$$

Finally, for $\lambda$ defined as in (20) we have

$$1 + \frac{1-\lambda}{\lambda} \cdot \ell = 1 + \left(1 - \frac{\ell(d-1)}{d(\ell-1)}\right)\frac{d(\ell-1)}{\ell(d-1)} \cdot \ell \qquad = 1 + \left(\frac{d(\ell-1)}{\ell(d-1)} - 1\right) \cdot \ell,$$

$$= 1 + \left(\frac{d(\ell-1) - \ell(d-1)}{\ell(d-1)}\right) \cdot \ell \qquad = 1 + \frac{\ell - d}{d-1} = \frac{\ell-1}{d-1},$$

which yields the desired formula for $\nu_d(\mathbf{u})$. $\qquad\square$

**On the norm of u.** The above theorem has a requirement on the norm of the vector $\mathbf{u}$ that defines the half-space $\mathcal{H}_\leq(\mathbf{u})$. However, in our cutting plane method we obtain $\mathbf{u}$ from Proposition 4.2 and Proposition 4.3, which do not have any guarantees on the norm of $\mathbf{u}$ explicitly. Crucially, at any given iteration $t$ of multidimensional backtracking with ellipsoids, we select a candidate preconditioner $\mathbf{P}_t = \text{Diag}(\mathbf{p}_t)$ such that $\|\mathbf{p}_t\|_\mathbf{A} = 1/\sqrt{2d}$. Then, if it fails the Armijo condition in (4) and $\mathbf{u}_t$ is as given by Proposition 4.2, then we have $\mathbf{p}_t \in \mathcal{H}_>(\mathbf{u}_t)$, that is, the separating hyperplane excludes $\mathbf{p}_t$. As we will show, this implies that $\|\mathbf{u}\|_{\mathbf{A}^{-1}}$ is large.

**Lemma D.3.** *Let $\mathbf{A} \in \mathbb{R}^{d \times d}$ be positive definite and $\mathbf{p} \in \mathbb{R}^d_{\geq 0}$ be such that $\|\mathbf{p}\|_\mathbf{A} \leq \gamma$ for some $\gamma > 0$. Let $\mathbf{u} \in \mathbb{R}^d_{\geq 0}$ be such that $\mathbf{p} \in \mathcal{H}_>(\mathbf{u})$. Then $\|\mathbf{u}\|_{\mathbf{A}^{-1}} > 1/\gamma$.*

*Proof.* For the sake of contradiction, assume $\|\mathbf{u}\|_{\mathbf{A}^{-1}} \leq 1/\gamma$. Then $\|\mathbf{u}\|_{\mathbf{A}^{-1}} \cdot \|\mathbf{p}\|_\mathbf{A} \leq 1$. Thus, by the Cauchy-Schwartz inequality,

$$\langle \mathbf{u}, \mathbf{p} \rangle = \left\langle \mathbf{A}^{-1/2}\mathbf{u}, \mathbf{A}^{1/2}\mathbf{p} \right\rangle \leq \left\|\mathbf{A}^{-1/2}\mathbf{u}\right\| \cdot \left\|\mathbf{A}^{1/2}\mathbf{p}\right\| = \|\mathbf{u}\|_{\mathbf{A}^{-1}} \cdot \|\mathbf{p}\|_\mathbf{A} \leq 1.$$

This is a contradiction since $\mathbf{p} \in \mathcal{H}_>(\mathbf{u})$ and, therefore, $\langle \mathbf{u}, \mathbf{p} \rangle > 1$. $\qquad\square$

**On the volume decrease.** Although the formula $\nu_d(\mathbf{u})$ in Equation (21) can be hard to interpret, we show a simple bound when $\|\mathbf{u}\|^2_{\mathbf{A}^{-1}} \geq 2d$.

**Lemma D.4.** *Let $\mathbf{A} \in \mathbb{R}^{d \times d}$ be a positive definite matrix and $\mathbf{u} \in \mathbb{R}^d$ be such that $\|\mathbf{u}\|^2_{A^{-1}} > d$. For $c := d/\ell \in (0, 1)$ we have $\nu_d(\mathbf{u}) \leq \sqrt{c \cdot e^{1-c}}$, where $\nu_d$ is defined as in (21). In particular, if $\|\mathbf{u}\|^2_{A^{-1}} > d$, then $\nu_d(\mathbf{u}) \leq \sqrt[4]{e}/\sqrt{2}$.*

*Proof.* Define $\ell := \|\mathbf{u}\|^2_{\mathbf{A}^{-1}} > d$ and $c := d/\ell \in (0, 1)$. Then,

$$\nu_d(\mathbf{u})^2 = \left(\frac{d}{\ell}\right)^d \left(\frac{\ell-1}{d-1}\right)^{(d-1)} = \frac{d}{\ell} \cdot \left(\frac{d}{\ell} \cdot \frac{\ell-1}{d-1}\right)^{(d-1)} = c \cdot \left(c \cdot \frac{d/c - 1}{d-1}\right)^{(d-1)}$$

$$= c \cdot \left(\frac{d-c}{d-1}\right)^{(d-1)} = c \cdot \left(1 + \frac{1-c}{d-1}\right)^{(d-1)} \leq c \cdot e^{1-c},$$

where the last inequality follows since $1 + x \leq e^x$ for all $x \in \mathbb{R}$. In particular, note that $c \in (0, 1) \mapsto c \cdot e^{1-c}$ is increasing since the derivative of the mapping is positive on $(0, 1)$. Thus, if $\|\mathbf{u}\|_{\mathbf{A}^{-1}} \geq 2d$, then $c \leq \frac{1}{2}$ and $c \cdot e^{1-c} \leq (1/2) \cdot e^{1/2}$. $\qquad\square$

**Exploiting symmetry.** Let us now exploit symmetry to avoid using non-diagonal matrices in our ellipsoids. We use the notion of *axis-aligned* sets in the next few results. A set $\mathcal{X} \subseteq \mathbb{R}^d$ is *axis-aligned* if for any point $\mathbf{p} \in \mathcal{X}$, the reflections of $\mathbf{p}$ along the axes are also contained in $\mathcal{X}$. Formally, for any $\mathbf{s} \in \{\pm 1\}^d$, we have that if $\mathbf{p} \in \mathcal{X}$, then $\text{Diag}(\mathbf{s})\mathbf{p} \in \mathcal{X}$. Furthermore, with a slight abuse of notation define $\text{Diag}(\mathbf{A}) := \text{Diag}(\text{diag}(\mathbf{A}))$. That is, $\text{Diag}(\mathbf{A})$ is the diagonal matrix whose diagonal entries match those of $\mathbf{A}$. The idea is that the set $\{\mathbf{p} \in \mathbb{R}^d_{\geq 0} : \text{Diag}(\mathbf{p}) \text{ is valid}\}$ of diagonals of valid preconditioners is contained in the non-negative orthant. Yet, we can extend it by reflecting it over each of the axes. Although this may seem counter-intuitive, this translates the structure of our problem

into symmetry among all orthant, and this can be exploited elegantly. Formally, the set of diagonals of valid preconditioners reflected over each axis is given by set

$$\mathcal{P} := \{\, \mathbf{p} \in \mathbb{R}^d : \mathrm{Diag}(|\mathbf{p}|) \text{ is valid}\},$$

where $|\mathbf{p}|$ is the entry-wise absolute value of $\mathbf{p} \in \mathbb{R}^d$. The following lemma shows that when looking for low volume ellipsoids that contain an axis-aligned set, we can restrict out attention to axis-aligned ellipsoids, defined by a diagonal matrix. The following lemma can be seen as the polar statement of Goemans et al. (2009, Proposition 3.1), with the benefit of not requriring any matrix inversions.

**Lemma D.5.** *Let $\mathcal{X} \subset \mathbb{R}^d$ be an axis-aligned convex set and let $\mathbf{A} \in \mathbb{R}^{d \times d}$ be positive definite matrix such that $\mathcal{X} \subseteq \mathcal{E}(\mathbf{A})$. Then $\mathcal{X} \subseteq \mathcal{E}(\mathrm{Diag}(\mathbf{A}))$ and $\mathrm{Vol}(\mathcal{E}(\mathrm{Diag}(\mathbf{A}))) \le \mathrm{Vol}(\mathcal{E}(\mathbf{A}))$.*

*Proof.* Let us start by showing that $\mathcal{X} \subseteq \mathcal{E}(\mathrm{Diag}(\mathbf{A}))$. We use the notation $\mathrm{Diag}(\mathbf{v}) \cdot \mathcal{X}$ to denote the set $\mathrm{Diag}(\mathbf{v}) \cdot \mathcal{X} := \{\, \mathrm{Diag}(\mathbf{v}) \cdot \mathbf{x} : \mathbf{x} \in \mathcal{X} \}$. Since $\mathcal{X}$ is axis-aligned, we have

$$\mathcal{X} = \mathrm{Diag}(\mathbf{s}) \cdot \mathcal{X} \subseteq \mathrm{Diag}(\mathbf{s}) \cdot \mathcal{E}(\mathbf{A}) = \mathcal{E}(\mathrm{Diag}(\mathbf{s})\mathbf{A}\,\mathrm{Diag}(\mathbf{s})), \qquad \forall \mathbf{s} \in \{\pm 1\}^d.$$

Therefore, $\mathcal{X}$ is contained in each of the $2^d$ ellipsoids of the form $\mathcal{E}(\mathrm{Diag}(\mathbf{s})\mathbf{A}\,\mathrm{Diag}(\mathbf{s}))$. Thus,

$$\mathcal{X} \subseteq \bigcap_{\mathbf{s} \in \{\pm 1\}^d} \mathcal{E}(\mathrm{Diag}(\mathbf{s})\mathbf{A}\,\mathrm{Diag}(\mathbf{s})) \subseteq \mathcal{E}\left( \frac{1}{2^d} \sum_{\mathbf{s} \in \{\pm 1\}^d} \mathrm{Diag}(\mathbf{s})\mathbf{A}\,\mathrm{Diag}(\mathbf{s}) \right),$$

where the last inclusion follows since, for any set of positive definite matrices $\mathcal{M}$, one may verify that $\cap_{\mathbf{M} \in \mathcal{M}} \mathcal{E}(\mathbf{M}) \subseteq \mathcal{E}((1/|\mathcal{M}|) \sum_{\mathbf{M} \in \mathcal{M}} \mathbf{M})$. Finally, note that

$$\sum_{\mathbf{s} \in \{\pm 1\}^d} \mathrm{Diag}(\mathbf{s})\mathbf{A}\,\mathrm{Diag}(\mathbf{s}) = \mathrm{Diag}(\mathbf{A}).$$

Indeed, let $i, j \in \{1, \cdots, d\}$. If $i = j$, then $(\mathrm{Diag}(\mathbf{s})\mathbf{A}\,\mathrm{Diag}(\mathbf{s}))_{i,j} = \mathbf{A}_{i,j}$ for any $\mathbf{s} \in \{\pm 1\}^d$. If $i \ne j$, then

$$\sum_{\mathbf{s} \in \{\pm 1\}^d} (\mathrm{Diag}(\mathbf{s})\mathbf{A}\,\mathrm{Diag}(\mathbf{s}))_{i,j}$$

$$= \sum_{\mathbf{s} \in \{\pm 1\}^d :\, \mathbf{s}[i] \ne \mathbf{s}[j]} (\mathrm{Diag}(\mathbf{s})\mathbf{A}\,\mathrm{Diag}(\mathbf{s}))_{i,j} + \sum_{\mathbf{s} \in \{\pm 1\}^d :\, \mathbf{s}[i] = \mathbf{s}[j]} (\mathrm{Diag}(\mathbf{s})\mathbf{A}\,\mathrm{Diag}(\mathbf{s}))_{i,j}$$

$$= 2^{d-1} \cdot (-\mathbf{A}_{i,j}) + 2^{d-1} \cdot \mathbf{A}_{i,j} = 0.$$

Let us now show that $\mathrm{Vol}(\mathcal{E}(\mathrm{Diag}(\mathbf{A}))) \le \mathrm{Vol}(\mathcal{E}(\mathbf{A}))$. Note that $\log(\mathrm{Vol}(\mathcal{E}(\mathbf{A}))) = \log(\mathrm{Vol}(\mathcal{E}(\mathbf{I}))) - \frac{1}{2} \log \det(\mathbf{A})$. Since $\log \det(\cdot)$ is concave over positive definite matrices, we have

$$\log \det(\mathrm{Diag}(\mathbf{A})) = \log \det\left( \frac{1}{2^d} \sum_{\mathbf{s} \in \{\pm 1\}^d} \mathrm{Diag}(\mathbf{s})\mathbf{A}\,\mathrm{Diag}(\mathbf{s}) \right)$$

$$\ge \frac{1}{2^d} \sum_{\mathbf{s} \in \{\pm 1\}^d} \log \det\left( \mathrm{Diag}(\mathbf{s})\mathbf{A}\,\mathrm{Diag}(\mathbf{s}) \right) = \frac{1}{2^d} \cdot 2^d \log \det(\mathbf{A}) = \log \det(\mathbf{A}).$$

Therefore,

$$\log(\mathrm{Vol}(\mathcal{E}(\mathrm{Diag}(\mathbf{A})))) = \log(\mathrm{Vol}(\mathcal{E}(\mathbf{I}))) - \tfrac{1}{2} \log \det(\mathrm{Diag}(\mathbf{A}))$$
$$\le \log(\mathrm{Vol}(\mathcal{E}(\mathbf{I}))) - \tfrac{1}{2} \log \det(\mathbf{A})$$
$$= \log(\mathrm{Vol}(\mathcal{E}(\mathbf{A}))),$$

which implies that $\mathrm{Vol}(\mathcal{E}(\mathrm{Diag}(\mathbf{A}))) \le \mathrm{Vol}(\mathcal{E}(\mathbf{A}))$. $\qquad \square$

We are now in position to prove Lemma 5.2, which follows directly from the previous two results.

*Proof of Lemma 5.2.* By the assumptions in Proposition 4.2 we have that $\mathbf{P} := \mathrm{Diag}(\mathbf{p})$ fails the Armijo condition 4 condition and, thus, $\mathbf{p} \in \mathcal{H}_>(\mathbf{u})$. This together with the assumption that $\|\mathbf{p}\|_{\mathbf{A}} \le 1/\sqrt{2d}$ imply via Lemma D.3 that $\|\mathbf{u}\|_{\mathbf{A}^{-1}} \ge \sqrt{2d}$. This allows us to use Theorem D.2 to find a new ellipsoid containing $\mathcal{E}(\mathbf{a}) \cap \mathcal{H}_\le(\mathbf{u})$ with the required volume decrease by Lemma D.4.

Yet, this ellipsoid may not be axis-aligned. We shall exploit the symmetry described in Lemma D.5 to show that the axis-aligned ellipsoid $\mathcal{E}(\mathbf{a}^+(\mathbf{a}, \mathbf{u}))$ enjoys the same guarantees.

Formally, we need $\mathcal{E}(\mathbf{a}^+(\mathbf{a}, \mathbf{u}))$ to contain $\mathcal{E}(\mathbf{a}) \cap \mathcal{H}_{\leq}(\mathbf{u})$. Since $\mathbf{u} \geq 0$, we have

$$\mathcal{H}_{\leq}(\mathbf{u}) \subseteq \mathcal{S}(\mathbf{u}) \coloneqq \{\, \mathbf{p} \in \mathbb{R} : \mathrm{Diag}(\mathbf{s}) \cdot \mathbf{p} \in \mathcal{H}_{\leq}(\mathbf{u}) \text{ for all } \mathbf{s} \in \{\pm 1\}^d \}.$$

Thus, it suffices for $\mathcal{E}(\mathbf{a}^+(\mathbf{a}, \mathbf{u}))$ to contain $\mathcal{E}(\mathbf{a}) \cap \mathcal{S}(\mathbf{u})$. From Theorem D.2 we know that $\mathcal{E}(\mathbf{a}) \cap \mathcal{S}(\mathbf{u})$ is contained in the ellipsoid given by the matrix $\lambda \mathrm{Diag}(\mathbf{a}) + (1 - \lambda)\mathbf{u}\mathbf{u}^{\mathsf{T}}$ for any $\lambda$, in particular for $\lambda$ as in (20) since $\|\mathbf{u}\|_{\mathbf{A}^{-1}} > \sqrt{d}$. Since $\mathcal{S}(\mathbf{u})$ is axis-aligned, we can exploit symmetry using Lemma D.5, which tells that $\mathcal{E}(\mathbf{a}) \cap \mathcal{S}(\mathbf{u})$ is contained in the ellipsoid given by the matrix

$$\mathrm{Diag}\left(\lambda \mathrm{Diag}(\mathbf{a}) + (1 - \lambda)\mathbf{u}\mathbf{u}^{\mathsf{T}}\right) = \mathrm{Diag}(\mathbf{a}^+(\mathbf{a}, \mathbf{u})),$$

as desired. Finally, the bound on the volume follows by Theorem D.2 and the bound on $\nu_d(\mathbf{u})$ given by Lemma D.4 since $\|\mathbf{u}\|_{\mathbf{A}^{-1}} \geq \sqrt{2d}$. $\qquad\square$

Finally, we are in position to prove Theorem 5.3, which follows almost directly from Lemma 5.2.

*Proof of Theorem 5.3.* Note that (a) holds by induction and since, by Proposition 4.2, we have $\mathrm{diag}(\mathbf{P}^*) \in \mathcal{H}_{\leq}(\mathbf{u}_t)$ for any $\mathbf{u}_t$ used in a call to CUT. For (b), fix $t \in \{1, \ldots, T\}$ and recall that in this case we have $\mathcal{S}_t = \{\, \mathrm{Diag}(\mathbf{p}) : \mathbf{p} \in \mathcal{E}(\mathbf{a}_t) \}$. As described in (8), one may verify that $\mathrm{Diag}(\mathbf{q}_t^*)$ for $\mathbf{q}_t^*$ given by

$$\mathbf{q}_t^* \coloneqq \frac{1}{\|\nabla f(\mathbf{x}_t)^2\|_{\mathbf{A}_t^{-1}}} \cdot \mathbf{A}_t^{-1} \nabla f(\mathbf{x}_t)^2$$

maximizes $\|\nabla f(\mathbf{x}_t)\|_{\mathbf{P}}$ for $\mathbf{P} \in \mathcal{S}_t$. Since

$$\mathbf{P}_t = \text{CANDIDATE}(\mathcal{S}_t, 1/\sqrt{2d}, \mathbf{x}_t) = \frac{1}{\sqrt{2d}} \mathrm{Diag}(\mathbf{q}_t^*),$$

we conclude that $\mathbf{P}_t$ is $1/\sqrt{2d}$-competitive. For (c), first note that we may assume $(1/L)\mathbb{1} \in \mathcal{E}(\alpha_0 I)$. To see that, assume $(1/L)\mathbb{1} \notin \mathcal{E}(\alpha_0 I)$, implying $\alpha_0 d > L^2$. In this case, any candidate preconditioner computed by CANDIDATE is always valid and, thus, we never call CUT. To see this, let $\mathbf{A}_0 \coloneqq \alpha_0 \mathbf{I}$ be the matrix defining the initial ellipsoid. Then, by the definition of CANDIDATE for ellipsoids we have that $\mathbf{P}_0 = \mathrm{Diag}(\mathbf{p}_0)$ is such that

$$\|\mathbf{p}_0\|_{\mathbf{A}_0}^2 = \alpha_0 \|\mathbf{p}_0\|^2 = \frac{1}{2d} < \frac{1}{2}\frac{\alpha_0}{L^2}.$$

Therefore, $\mathbf{p}_0[i] \leq 1/L$ for all $i \in \{1, \ldots, d\}$, which implies that $\mathbf{P}_0$ is valid since $\mathbf{P}_0 \preceq \frac{1}{L}\mathbf{I}$.

Let us look now at the case $(1/L)\mathbb{1} \in \mathcal{E}(\alpha_0 I)$. Therefore, $\mathcal{B}(1/L\mathbb{1}) \subseteq \mathcal{E}(\mathbf{a}_t)$ for all iterations $t$. Since the minimum volume ellipsoid containing the box $\mathcal{B}((1/L)\mathbb{1})$ is the unit sphere of radius $1/L$, that is, $\mathcal{E}((L^2/d)\mathbb{1})$. Therefore, $\mathrm{Vol}(\mathcal{E}(\mathbf{a}_t)) \geq \mathrm{Vol}(\mathcal{E}((L^2/d)\mathbb{1}))$. Moreover, every time we call cut the volume of the ellipsoid goes down by $1/c \coloneqq \sqrt[4]{e}/\sqrt{2}$. Therefore, the total number of calls to CUT is no more than

$$\log_c\left(\frac{\mathrm{Vol}(\mathcal{E}(\alpha_0\mathbb{1}))}{\mathrm{Vol}(\mathcal{E}((L^2/d)\mathbb{1}))}\right) = \log_c\left(\frac{L^d}{d\alpha_0^{d/2}}\right) \leq \frac{d}{\log(c)}\log\left(\frac{L}{d\alpha_0}\right) \leq 12d\log\left(\frac{L}{\alpha_0}\right)$$

since $\log(c) \geq 1/12$. $\qquad\square$

**Refining the choice of $\lambda$.** Although we have shown in Lemma 5.2 a choice a $\lambda$ that guarantees volume decrease, it may be sub-optimal. The choice of $\lambda$ in Equation (20) is inherited from the non-symmetric case in Theorem D.2. Although Lemma 5.2 and Theorem D.2 match when $\mathbf{u}$ has only one non-zero entry, we should expect better choices of $\lambda$, leading to more volume shrinkage, to be possible in Lemma 5.2. Although we have not found a choice of $\lambda$ that is dependent on $\mathbf{u}$ that generically improves upon (20), in practice we can solve for a better $\lambda$ numerically, by directly minimizing the volume of the resulting ellipsoid,

$$\min_{0 < \lambda < 1} \mathrm{Vol}(\mathcal{E}(\lambda \mathbf{a} + (1 - \lambda)\mathrm{Diag}(\mathbf{u}\mathbf{u}^{\top}))) = \min_{0 < \lambda < 1} -\sum_i \log(\lambda \mathbf{a}[i] + (1 - \lambda)\mathbf{u}[i]^2).$$

As the problem is one-dimensional, numerical solvers can often find near-optimal solutions. By warm-starting a numerical solver with the $\lambda$ defined in (20), we can guarantee that the resulting ellipsoid leads to a smaller volume and we do not lose our worst-case theoretical guarantees.

**Connection to the polar problem and related work.** Our results have an interesting connection to some of the results from Goemans et al. (2009), via the use of polarity theory. Here we give a quick overview of their work and the connection to our cutting plane methods. Goemans et al. (2009) shows techniques to approximate some polyhedron $\mathcal{P} \subseteq \mathbb{R}^d$ (a polymatroid being one of the main examples) *from inside* by some ellipsoid $\mathcal{E}(\mathbf{A})$. Their algorithm maintains an ellipsoid $\mathcal{E}(\mathbf{A}) \subseteq \mathcal{P}$ and tries to iteratively enlarge it. They assume access to an oracle such that, at each iteration, either finds a point $\mathbf{u} \in \mathbf{P}$ that is sufficiently far from $\mathcal{E}(\mathbf{A})$, meaning $\|\mathbf{u}\|_{\mathbf{A}} > \sqrt{d} + \epsilon$ for some $\epsilon > 0$, or guarantees that $\mathcal{E}(\mathbf{A})$ "approximates well" $\mathcal{P}$ from inside in the sense that $\|\mathbf{u}\|_{\mathbf{A}} \leq (\sqrt{n} + \epsilon)/\alpha$ for all $\mathbf{u} \in \mathcal{P}$, where $\alpha > 0$ is some approximation factor. In their algorithm, when the oracle finds a point $\mathbf{u} \in \mathcal{P}$ such that $\|\mathbf{u}\|_{\mathbf{A}} > \sqrt{d} + \epsilon$ the algorithm needs to find an ellipsoid $\mathcal{E}(\mathbf{A}^+)$ such that

$$\mathcal{E}(\mathbf{A}^+) \subseteq \mathrm{conv}(\mathcal{E}(\mathbf{A}) \cup \{\mathbf{u}, -\mathbf{u}\}), \tag{22}$$

where $\mathrm{conv}(\mathcal{D})$ is the convex hull of $\mathcal{D}$. Interestingly, the polar problem is exactly what we need for out cutting plane method. More precisely, the polar set $\mathcal{X}^*$ of a set $\mathcal{X}$ is given by $\mathcal{X}^* := \{z \in \mathbb{R}^d : \langle z, x \rangle \leq 1\}$. Then, by taking polars and using that $\mathcal{E}(\mathbf{A})^* = \mathcal{E}(\mathbf{A}^{-1})$, we have that $\mathcal{P}^* \subseteq \mathcal{E}(\mathbf{A}^{-1})$. Moreover, taking polar on both sides of (22) yields that an equivalent problem is finding $(\mathbf{A}^+)^{-1}$ such that

$$\mathcal{E}((\mathbf{A}^+)^{-1}) \supseteq \mathcal{E}(\mathbf{A}^{-1}) \cap \{-\mathbf{u}, \mathbf{u}\}^* = \mathcal{E}(\mathbf{A}^{-1}) \cap \{\mathbf{z} : |\langle \mathbf{u}, \mathbf{z} \rangle| \leq 1\}.$$

That is, the problem is the one of finding a smaller ellipsoid $\mathcal{E}((\mathbf{A}^+)^{-1})$ that contains $\mathcal{E}(\mathbf{A}^{-1}) \cap \{\mathbf{z} : |\langle \mathbf{u}, \mathbf{z} \rangle| \leq 1\}$, which is broadly the goal of the subroutine CUT.

Table 1: **Datasets used in our experiments,** including number of samples $n$ and dimension $d$, and order of magnitude of the condition number of the regularized system ($\kappa(\mathbf{X}^\top\mathbf{X}+1/n\mathbf{I})$) and condition number of the system when using the optimal diagonal preconditioner, $\kappa_*$.

| Dataset | Repository/Source | $n$ | $d$ | $\kappa$ | $\kappa_*$ |
|---|---|---|---|---|---|
| **cpusmall** | LIBSVM, Delve (`comp-activ`) | 8 192 | 12 | $10^{13}$ | $10^2$ |
| **california-housing** | Scikit/StatLib, Kelley Pace and Barry (1997) | 20 640 | 8 | $10^{10}$ | $10^4$ |
| **power-plant** | UCI, Tüfekci (2014) | 9 568 | 4 | $10^9$ | $10^4$ |
| **concrete** | UCI, Yeh (1998) | 1 030 | 8 | $10^9$ | $10^3$ |
| **mg** | LIBSVM, Flake and Lawrence (2002) | 1 385 | 6 | $10^3$ | $10^3$ |
| **breast-cancer** | UCI | 569 | 32 | $10^{13}$ | $10^2$ |
| **australian** | LIBSVM, Statlog | 690 | 14 | $10^9$ | $10^2$ |
| **heart** | LIBSVM, Statlog | 270 | 13 | $10^7$ | $10^2$ |
| **diabetes** | UCI | 768 | 8 | $10^6$ | $10^2$ |
| **ionosphere** | UCI | 351 | 34 | $10^3$ | $10^2$ |
| **news20** | LIBSVM, Keerthi and DeCoste (2005) | 19 996 | 1 355 191 | $10^{13}$ | NA |
| **rcv1** | LIBSVM, Lewis et al. (2004) | 20 242 | 47 236 | $10^{13}$ | NA |

## E  Experiments

### Objective functions

We use $L_2$-regularized linear regression $\mathcal{L}_{\text{LINEAR}}$ and $L_2$-regularized logistic regression $\mathcal{L}_{\text{LOGISTIC}}(\mathbf{w})$, with a regularization coefficient of 1. Given a data matrix $\mathbf{X} \in \mathbb{R}^{n\times d}$, target $y \in \mathbb{R}^n$ for regression tasks and $y \in \{0,1\}^n$ for classification tasks, and parameters $\mathbf{w} \in \mathbb{R}^d$,

$$\mathcal{L}_{\text{LINEAR}}(\mathbf{w}) = \frac{1}{n}\left(\frac{1}{2}\|\mathbf{X}\mathbf{w}-\mathbf{y}\|^2 + \frac{1}{2}\|\mathbf{w}\|^2\right).$$

$$\mathcal{L}_{\text{LOGISTIC}}(\mathbf{w}) = \frac{1}{n}\sum_{i=1}^{n}-\mathbf{y}[i]\log(\sigma(\langle\mathbf{x}_i,\mathbf{w}\rangle)) - (1-\mathbf{y}[i])\log(1-\sigma(\langle\mathbf{x}_i,\mathbf{w}\rangle)) + \frac{1}{n}\frac{1}{2}\|\mathbf{w}\|^2.$$

where $\mathbf{x}_i$ is the $i$th row of $\mathbf{X}$ and $\sigma$ is the sigmoid function, $\sigma(z) = 1/1+\exp(-z)$. For all datasets, we add a bias term by prepending a feature column of ones to $\mathbf{X}$.

### Datasets

We use the datasets listed in Table 1, made available by LIBSVM (Chang and Lin, 2011), Scikit-Learn (Pedregosa et al., 2011) and the UCI repository (Dua and Graff, 2017).

### Data rescaling

We do not rescale, standardize or otherwise change any of the datasets beyond adding a bias term, as our goal is to check whether preconditioned methods can handle badly scaled data.

### Initializations

We consider two types of initializations. The first approximates a "best-case" scenario where we start from an estimate with a reasonable loss value despite the bad scaling of the data. We set $\mathbf{w}[i] = 0$ except for the bias term $\mathbf{w}[0]$ which is set at the MLE of the non-regularized problem,

$$\mathbf{w}[0] = \bar{y} \qquad \text{where } \bar{y} = \tfrac{1}{n}\sum_{i=1}^{n}\mathbf{y}[i] \qquad \text{for linear regression,}$$

$$\mathbf{w}[0] = \log\left(\frac{\bar{y}}{1-\bar{y}}\right) \qquad \text{where } \bar{y} = \tfrac{1}{n}\sum_{i=1}^{n}\mathbf{y}[i] \qquad \text{for logistic regression.}$$

The results in the main text use this initialization. The second initialization takes $\mathbf{w} \sim \mathcal{N}(0,\mathbf{I})$, giving a starting point with potentially large loss. We give results using both initializations in the appendix.

**Optimizers used**

- For the small linear regression problems, we use preconditioned gradient descent with the optimal preconditioner, pre-computed using the semidefinite formulation of Qu et al. (2022), solved using CVXPY (Diamond and Boyd, 2016) based on the Matlab implementation of Qu et al.

  https://github.com/Gwzwpxz/opt_dpcond

- Gradient descent with a backtracking line-search with backtracking parameter $\gamma = 1/2$.

- RPROP (Riedmiller and Braun, 1993) following the implementation and default hyperparameters in PyTorch (Paszke et al., 2019) (starting step-size of $10^{-1}$, increase step-size factor $\eta^+ = 1.2$, decreasing step-size factor $\eta^- = 0.5$, minimum step-size of $10^{-6}$ and maximum step-size of 50).

  https://github.com/pytorch/pytorch/blob/v2.0.1/torch/optim/rprop.py

- Hypergradient descent to set the step-size, using (S)GD-HD (the multiplicative variant, Baydin et al., 2018). The hypergradient step-size is set to the default $\beta = 0.02$ (Baydin et al., 2018, footnote 3). The initial step-size is set to $\alpha_0 = 10^{-10}$, as otherwise most runs diverged immediately.

- The diagonal Barzilai-Borwein method of Park et al. (2020), using their non-monotonic line-search. We use the default parameters suggested; a starting step-size of $10^{-6}$, regularization factor on the previous diagonal approximation $\mu = 10^{-6}$, a backtracking factor of $1/2$ for the backtracking line-search and a window of 15 steps for the non-monotone line-search. This line-search does not use a forward step as the update can increase the preconditioner.

- Preconditioned gradient descent using the diagonal Hessian, with a backtracking line-search.

- AdaGrad (Duchi et al., 2011) but augmented with a backtracking line-search as suggested by Vaswani et al. (2020) to make it competitive in the deterministic setting, following the PyTorch (Paszke et al., 2019) implementation.

  https://github.com/pytorch/pytorch/blob/v2.0.1/torch/optim/adagrad.py

**Line-search and forward steps**

For all methods, the backtracking line-search is augmented by a forward step. When a step-size is accepted, it is increased by a factor of $1.1$ for the next step. For multidimensional backtracking, we increase the set uniformly, taking $\mathbf{b}' = 1.1 \cdot \mathbf{b}$ for the box and $\mathbf{a}' = \mathbf{a}/\sqrt{1.1}$ for the ellipsoid. The ellipsoid uses a slightly smaller increase factor.[7]

**Hyperparameters for the line-search and multidimensional backtracking**

For the backtracking line-searches used in gradient descent, preconditioned gradient descent and used to augment the other algorithms, we start the search at an initial step-size of $10^{10}$ and backtrack by a factor of $1/2$ when failing the Armijo condition, implemented generically as

$$f(\mathbf{x} - \mathbf{d}) \leq f(\mathbf{x}) - \tfrac{1}{2}\langle \nabla f(\mathbf{x}), \mathbf{d} \rangle$$

For multidimensional backtracking, we initialize the sets such that the first preconditioner is on the order of $10^{10}\mathbf{I}$. Using the notation of Appendix A, we use the scaling factor $c_0 = d \cdot 10^{10}$ for the box variant and $c_0 = \sqrt{d} \cdot 10^{10}$ for the ellipsoid variant. The first preconditioner tried by the box variant with backtracking factor $\gamma = 1/2d$ is then $1/2 \cdot 10^{10}\mathbf{I}$, and the first preconditioner tried by the ellipsoid variant (assuming the gradient is uniform, $\nabla f(\mathbf{x}_0) \propto \mathbf{1}$) is $1/\sqrt{2} \cdot 10^{10}\mathbf{I}$.

---

[7]To increase by a factor of $1.1$ in the one-dimensional case, the update to the ellipsoid should be $\mathbf{a}' = \mathbf{a}/1.1^2$.

Table 2: Running times for parts of a backtracking update using the Ellipsoid variant (Figure 11) on RCV1. Average runtime over 100 calls, $\pm$ standard deviations over 10 repeats.

| Operation | Average runtime | $\pm$ std |
|---|---|---|
| Compute gradient, preconditioner and next iterate | 24.4 ms | $\pm$0.2 ms |
| Compute hypergradient | 12.6 ms | $\pm$0.1 ms |
| Compute CUT (using Lemma 5.2) | 0.9 ms | $\pm$0.1 ms |
| Compute CUT (solving the convex combination with `scipy.optimize`) | 7.6 ms | $\pm$0.1 ms |

## E.1 Performance comparison

The algorithms we compare do not have a unified notion of an "iteration". For example, a backtracking line-search and multidimensional backtracking do more work than plain gradient descent in between updates to the parameters. To make the performance comparison fair, the results in Figures 1 and 5 account for the overhead of backtracking by comparing the performance against the number of oracle calls, counting the number function and gradient evaluations.

This metric is a good proxy for the work required by multidimensional backtracking. The majority of the computation cost of the "backtracking" part of the algorithm comes from computing the gradient at the next point (to compute the hypergradient). The overhead due to other operations such as the CUT method are minimal compared to gradient computations. Indeed, beyond the gradient computation, multidimensional backtracking only involves a few vector operations (see Figures 10 and 11). Even solving for the best convex combination numerically to obtain tighter ellipsoids (as in the CUT method in Figure 11) is faster than computing gradients. To illustrate this point, we provide the running times for subsets of a backtracking update from multidimensional backtracking in Table 2.

## E.2 Additional results

Figures 15–20 give additional results on small linear and logistic regression problems and large logistic regression problems. Multidimensional backtracking has a consistent performance across problems and does not suffer from the extremely bad conditioning of cpusmall or california-housing (linear regression) or australian, breast-cancer, diabetes and heart (logistic regression).

Figure 15: Runs on small linear regression datasets with Bias initialization

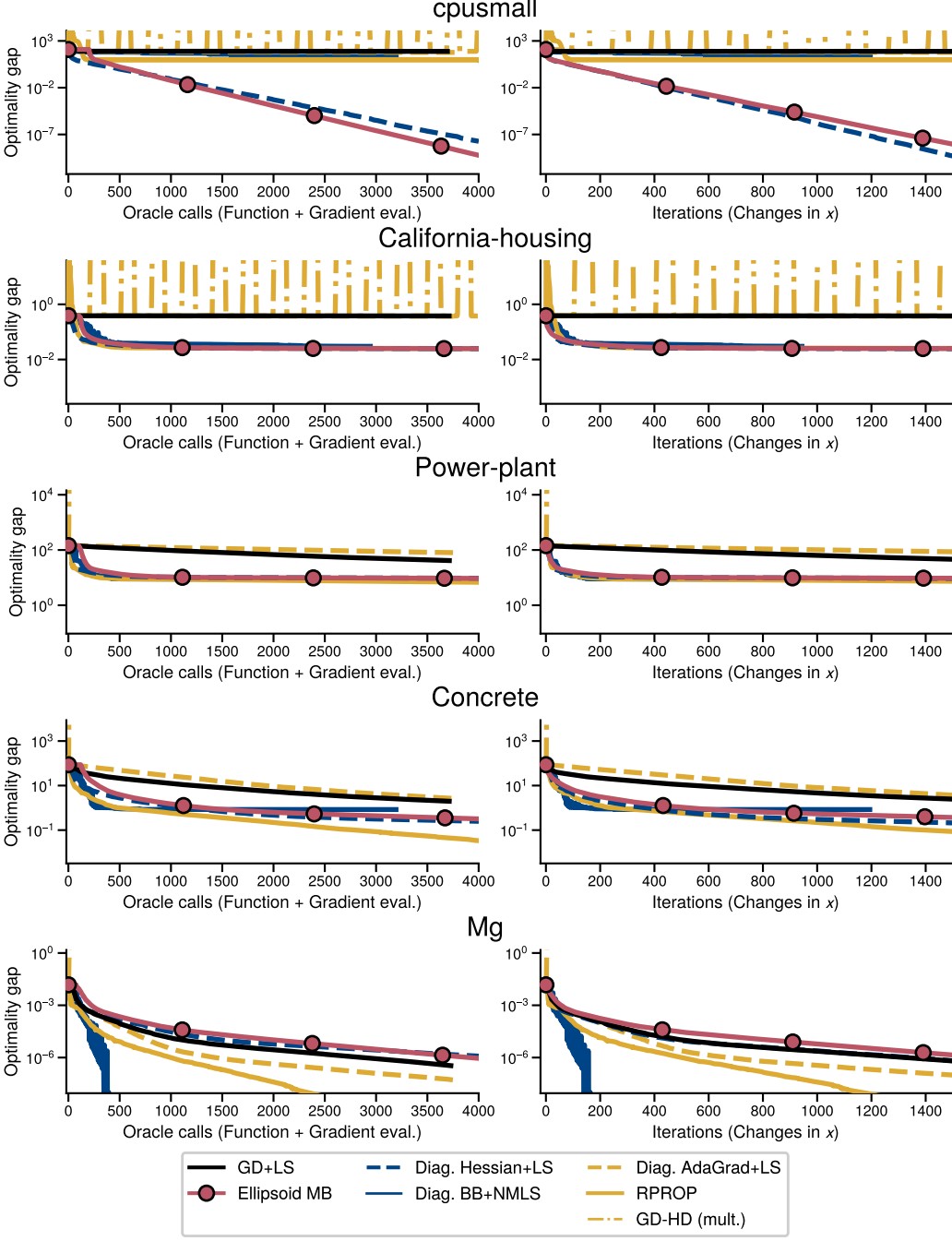

Figure 16: Runs on small linear regression datasets with Gaussian initialization

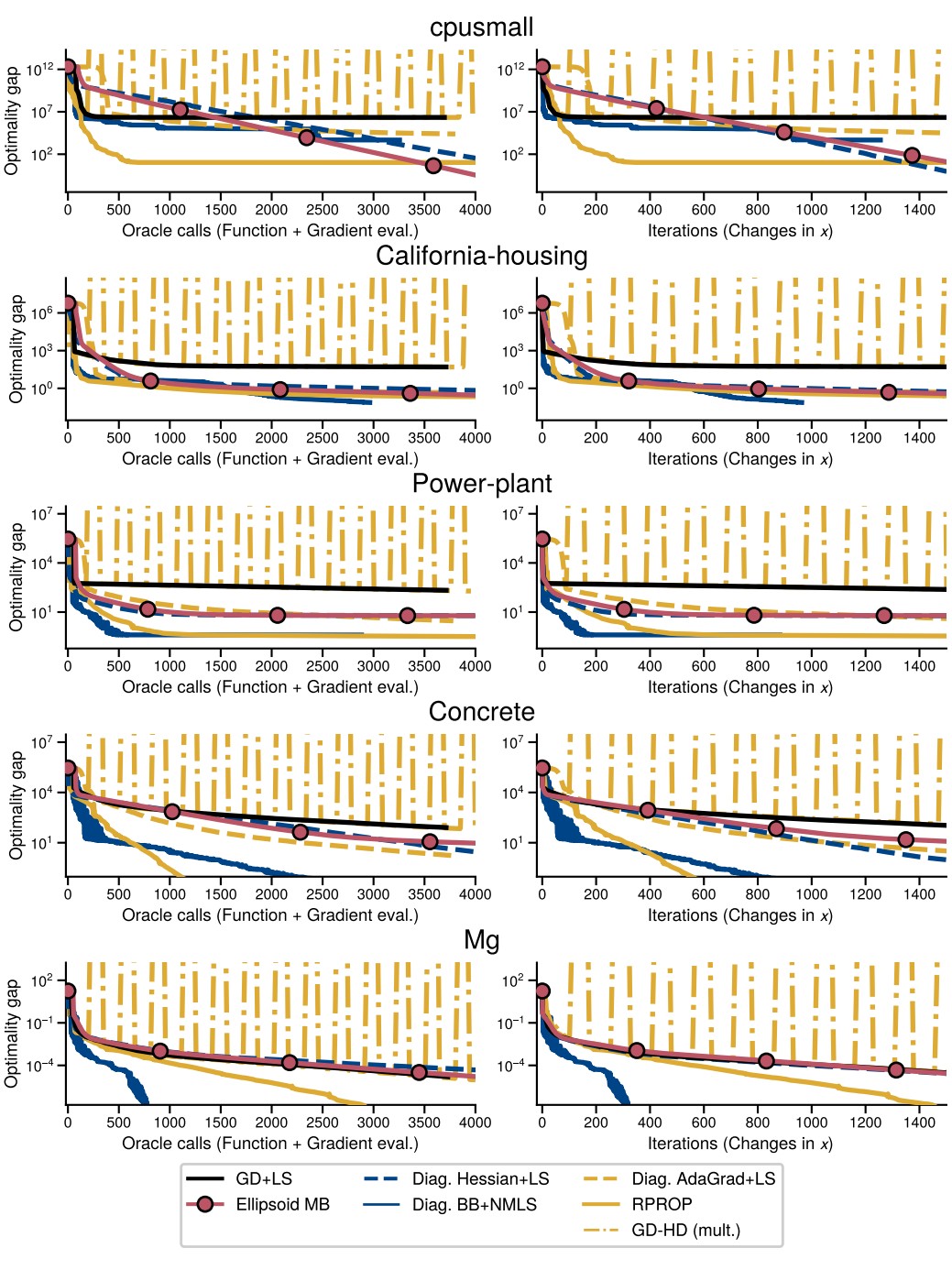

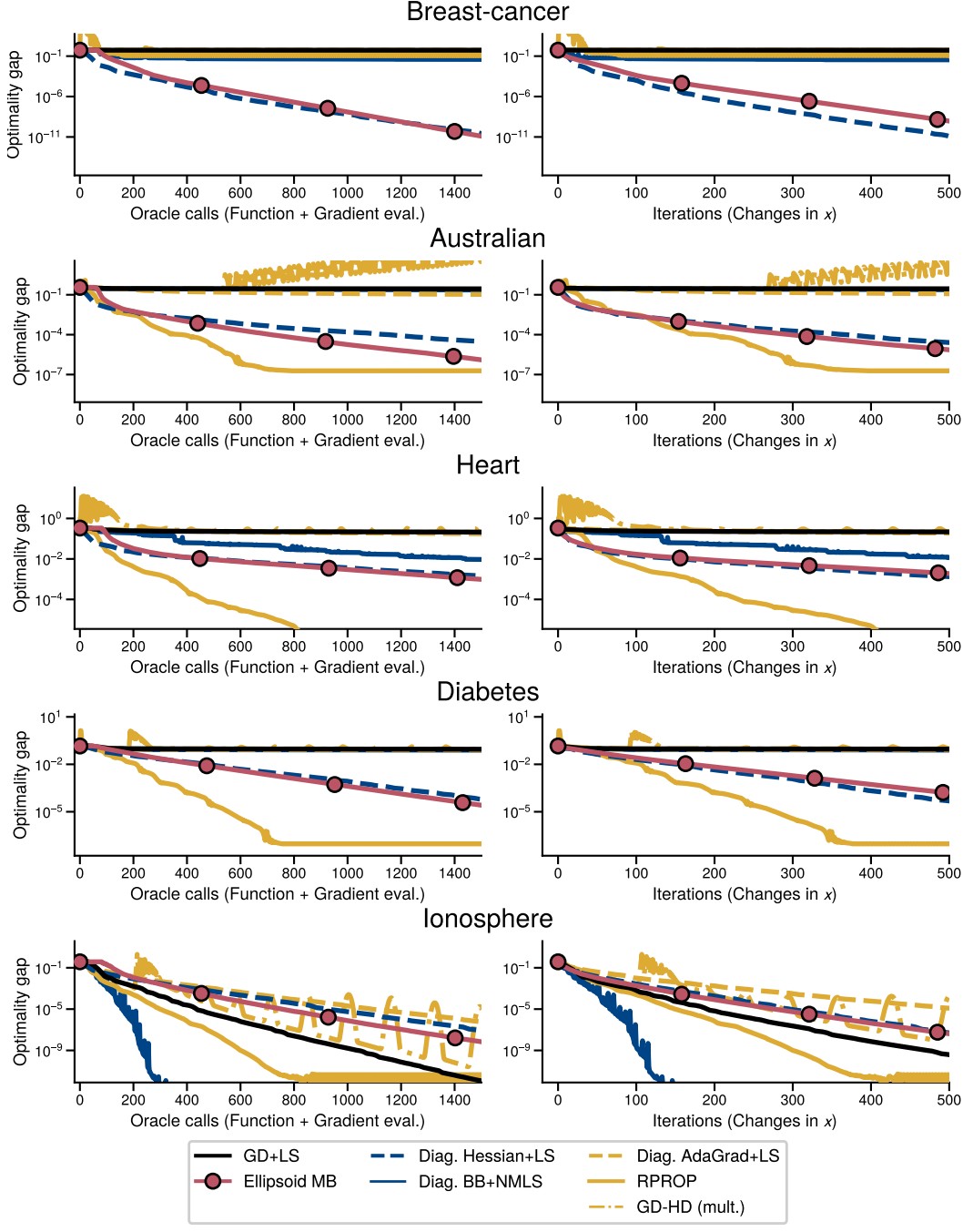

Figure 17: Runs on small logistic regression datasets with Bias initialization

Figure 18: Runs on small logistic regression datasets with Gaussian initialization

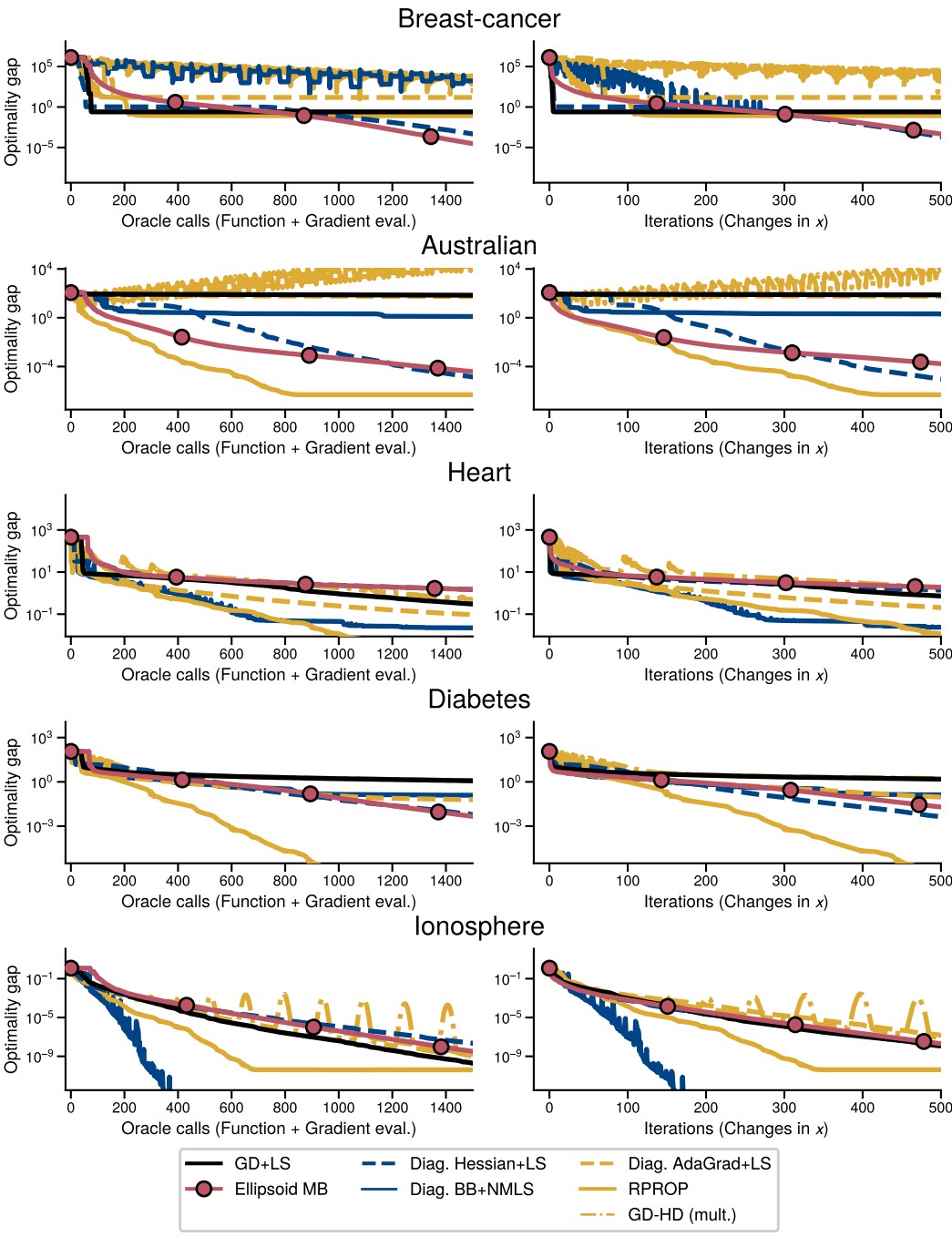

Figure 19: Runs on large logistic regression datasets with Bias initialization

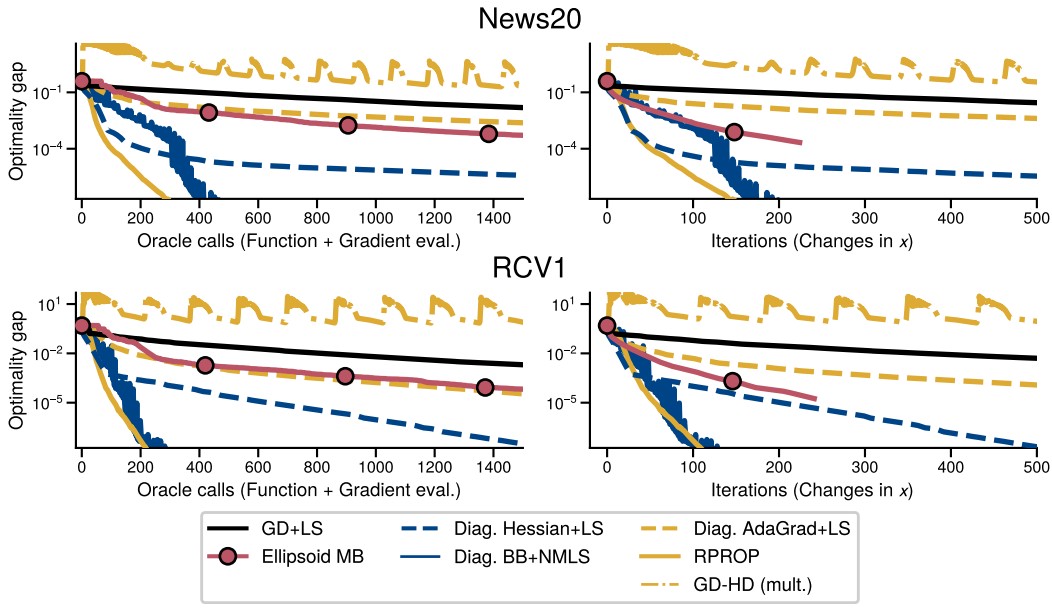

Figure 20: Runs on large logistic regression datasets with Gaussian initialization

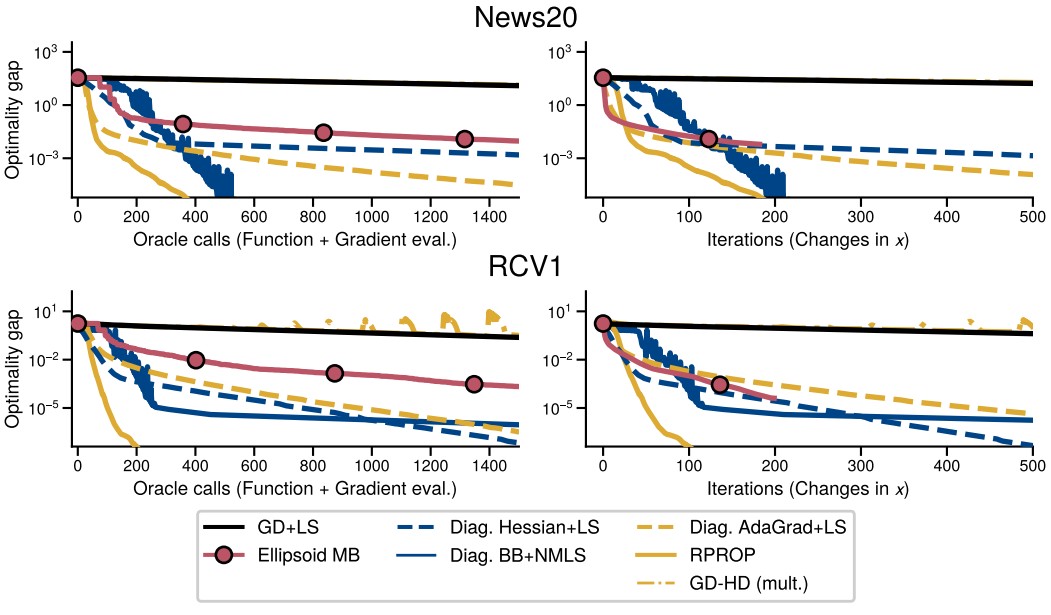

