# OpenReview forum: "Searching for Optimal Per-Coordinate Step-sizes with Multidimensional Backtracking"
_NeurIPS.cc/2023/Conference — NeurIPS 2023 poster_

### Official Review · Reviewer_eKZX · 2023-07-06

**Soundness:** 3 good
**Presentation:** 2 fair
**Contribution:** 2 fair
**Rating:** 5
**Confidence:** 1

**Summary:**

The authors suggest incremental updates of $\mathbf{x}$ for finding the minimum of strongly-convex function $f$ that guarantee decreasing $f(\mathbf{x}_{t})-f(\mathbf{x}_\ast)$ based on only 1st-order gradient information.

Their idea is in each step,
- choose a candidate matrix $\mathbf{P}_t$ based on set $\mathcal{S}_t$, and
- check the condition (4), that guarantee sufficient decrease of the $f$, and
- if the condition is satisfied:
    - apply update $\mathbf{x}_{t+1} = \mathbf{x}_t - \mathbf{P}_t\nabla f(\mathbf{x}_t)$
    - $\mathcal{S}_{t+1} = \mathcal{S}_t$
- else
    - $\mathbf{x}_{t+1} = \mathbf{x}_t$
    - update $\mathcal{S}_{t+1} = \text{cut}(\mathcal{S}_t, \mathbf{x}_t, \mathbf{P}_t)$

They provide proofs for
- approaching the optimal in Proposition 3.2
- how to choose candidate and cut algorithm in Theorem 5.3, and its maximum number of calls.

They conduct some simple experiments in Section 6, and show the proposed algorithm's efficiency.

**Strengths:**

originality
- considering backtracking using preconditioned matrix $\mathbf{P}_t$ would be novel idea. But I'm not an expert of this field, and not so sure on the originality.

quality
- The proposed algorithm is supported by some proofs, and it shows good empirical results.

clarity
- Basically, the manuscript is readable.

significance
- The proposed algorithm seems be better than other baselines except for Diag. Hessian+LS, which uses information of 2nd order derivatives, i.e. Hessian, even the proposed algorithm uses 1st-order derivatives of $f$. It would be significant.

**Weaknesses:**

- The manuscript contains some typos.
- I have a concern on Figure 5. The horizontal axis shows number of f/grad evals, but I guess the number of CUT calls should be also taken into account.


**Questions:**

- In line 129, the representation $\nabla^2 f$ appears. Does it mean Hessian? or Laplacian?
- Between line 197 and 198, the final inequality may be not $\overset{(3)}{\leq}$ but $\overset{(4)}{\leq}$ ?
- In the caption of Figure 3(b), $\mathcal{H}\_{>}(\mathbf{u})$ should be replaced by $\mathcal{H}_>(\mathbf{v})$ ?
- In Figure 5, do the authors take the number of CUT into account?

**Limitations:**

They address the limitation, their method is only supported for convex deterministic setting.

---

> ### Author Rebuttal · Authors · 2023-08-10
>
> Thank you for engaging with our paper during the review period!
>
> Please see the discussion of the overhead of `CUT` (also raised by reivewer JVBS) in the overall response. For the other points:
>
> > considering backtracking using preconditioned matrix $\mathbf{P}_t$ would be novel idea. But I'm not an expert of this field, and not so sure on the originality.
>
> We are not aware of prior work using similar approaches to estimate a preconditioner, or provide guarantees similar to the ones we present. We emphasize that, beyond the specific algorithm presented, a contribution of our work is the development of a formal definition of adaptive per-coordinate step-sizes and a way to provably find them.
>
> As mentioned in the introduction, existing definitions of adaptivity do not capture the benefits of preconditioning, even on simple linear regression problems. For example, the online learning definition used by AdaGrad forces the step-size to go to 0 and leads to poor performance in practice.
>
> We believe that these ideas can lead to further work to improve adaptive methods.
>
> > In line 129, the representation $\nabla^2 f(x)$ appears. Does it mean Hessian? or Laplacian?
>
> $\nabla^2 f$ does refer to the Hessian. We will mention it on first appearance to avoid confusion.
>
> > Between line 197 and 198, the final inequality may be not (3) but (4)?
>
> The numbers on the display math do not refer to equation numbers (which we assume is the confusion) but to the numbers `(1), (2), (3)` in the preceding paragraph. We will change those to `(a), (b), (c)` to disambiguate.
>
> > In the caption of Figure 3(b), $\\mathcal{H}\_{>}(\\mathbf{u})$ should be replaced by $\\mathcal{H}\_{>}(\\mathbf{v})$?
>
> Indeed, we will fix it.

---

### Official Review · Reviewer_hZtF · 2023-07-09

**Soundness:** 4 excellent
**Presentation:** 4 excellent
**Contribution:** 4 excellent
**Rating:** 8
**Confidence:** 4

**Summary:**

This paper provides a backtracking approach for smooth convex optimization on a per-coordinate basis with a theoretical analysis that show the gain with respect to classical backtracking line-search and that compare to the optimal per-coordinate conditioners.

**Strengths:**

This paper is super well written and organized. The contribution is also significant as it is a building block of many problems in machine learning.
In general, further improving the "adaptivity" of optimization algorithms is essential to seamlessly apply theoretical results (i.e., optimal per coordinate step sizes) to operational purposes.

**Weaknesses:**

The only drawback might be focusing on smooth and strongly convex problems, but it is still a significant first step.


**Questions:**

None.

**Limitations:**

None.

---

> ### Author Rebuttal · Authors · 2023-08-10
>
> Thank you for sharing in our excitement with the paper!
>
> We agree that one of the major drawbacks of our technique is the focus on the smooth, strongly-convex case. We do address the PL case as a relaxation of strong-convexity in the Appendix, and hope our work will lead to others exploring relaxations like the convex-only, non-convex, stochastic, and other cases.

---

### Official Review · Reviewer_5i17 · 2023-07-15

**Soundness:** 3 good
**Presentation:** 2 fair
**Contribution:** 3 good
**Rating:** 6
**Confidence:** 1

**Summary:**

This paper extends backtracking to multi-dimension. The authors propose a cutting plane method to find optimal per-coordinate step-sizes (in other words, to find an optimal preconditioner) for smooth convex optimisation. Experiments on ill-conditioned logistic regression problems show that the proposed algorithm can find good preconditioner and improve over vanilla gradient descent.

**Strengths:**

This paper fills a potential gap in the optimization literature by proposing multidimensional backtracking. The proposed method is technically sound and seems to work well in practice.

**Weaknesses:**

I do not see any major issues with the paper, except maybe that it is a bit hard to follow and understand (even though the English is good). Maybe because I don't have enough background on the topic. I'm really sorry for the short review.

**Questions:**

None.

**Limitations:**

See weaknesses.

---

> ### Author Rebuttal · Authors · 2023-08-10
>
>
> Thank you for engaging with our paper during the review period!
>
> > This paper fills a potential gap in the optimization literature by proposing multidimensional backtracking
>
> We emphasize that, beyond the specific algorithm presented, a contribution of our work is the development of a formal definition of adaptive per-coordinate step-sizes and a way to provably find them.
>
> As mentioned in the introduction, existing definitions of adaptivity do not capture the benefits of preconditioning, even on simple linear regression problems. For example, the online learning definition used by AdaGrad forces the step-size to go to 0 and leads to poor performance in practice.
>
> We believe that these ideas can lead to further work to improve adaptive methods.

---

### Official Review · Reviewer_JVBS · 2023-07-21

**Soundness:** 4 excellent
**Presentation:** 4 excellent
**Contribution:** 4 excellent
**Rating:** 8
**Confidence:** 4

**Summary:**

This paper presents a generalized backtracking line-search method, which estimates coordinate-wise stepsizes referred to as 'preconditioner' of gradient descent. Stemmed from the observation that any existing methods do not exceed the performance of backtracking line-search method, this paper designs a generalized backtracking line-search technique which is realized as a cutting plane method, whose separating hyperplane comes from the hypergradient, i.e., gradient with respect to hyperparameter of the algorithm, which is a stepsize in this case. Followed by the worst-case convergence analysis for smooth strongly convex function, the writers also provide experimental results illustrating the competitiveness of this method for ill-conditioned problems and robustness among problem classes.


**Strengths:**

Section 4 contains the key insight of this work: that a failed preconditioner (defined as one that violates a Armijo-type condition) provides a cutting plane on the set of valid preconditioners. This is a very nice idea that is, as far as I know, novel, and I expect this work to lead to a lot of follow-up work. This is a new type of result and I think it is valuable.

**Weaknesses:**

.

**Questions:**

The 'CUT' subroutine, which is a 'backtracking' phase of this algorithm, can be called up to the number of iterations linear in dimension $d$, which can be large in considering large-scale problems. It has been illustrated in the experimental result that for large problems it recovers preconditioner quite fast, but I'm also curious on how much total overhead is caused by the subroutine `CUT'.


 (p.3 line 121) It seems there is a typo on notation regarding $d, n, \\alpha$.


**Limitations:**

.

---

> ### Author Rebuttal · Authors · 2023-08-10
>
> Thank you for sharing in our excitement with the paper!
>
> Please see the discussion of the overhead of `CUT` (also raised by reivewer eKZX) in the overall response.
>
> Thanks you for spotting the typo on l.121, a sentence got eaten due to a version conflict.

---

### Author Rebuttal · Authors · 2023-08-10


We thank all reviewers for engaging with our paper. We were very pleased to see reviewers JVBS and hZtF sharing in our excitement with the paper and appreciate the great feedback. We appreciate that reviewers 5i17 and eKZX truly engaged with our paper despite it being outside of their area of expertise.

If the reviewers have more feedback about which parts would benefit from more exposition to improve the presentation reach a broader audience, we would appreciate specific recommendations. This feedback would be valuable for future expositions of our work.

### On the overhead of CUT

Reviewers eKZX and JVBS both asked for more details on the overhead of backtracking and the `CUT` operation. The results in Figure 1 and 5 account for the overhead of backtracking by showing the number of gradient evaluations. The majority of the computation cost of a backtracking step comes from computing the gradient at next point to compute the (hyper-)gradient. This is why our algorithm does not make progress at the start, as the first gradient evaluations are spent on backtracking.

The overhead of `CUT` is minimal compared to gradient computations as it only involves a few vector operations (see Figure 11 in Appendix A for the pseudocode). Even solving for the best convex combination numerically (see lines 287-288 or Appendix D) is faster than computing gradients. The table below gives running times fors parts of a backtracking update on RCV1 for the Ellipsoid version (average runtime over 100 calls, ±std over 10 repeats).

| Operation                                                                | Average runtime | ±std    |
|--------------------------------------------------------------------------|-----------------|---------|
| Compute gradient, preconditioner and next iterate                                                     | 24.4 ms         | ±0.2 ms |
| Compute hypergradient                                                    | 12.6 ms         | ±0.1 ms |
| Compute `CUT` (using Lemma 5.2)                                          |  0.9 ms         | ±0.1 ms |
| Compute `CUT` (using `scipy.optimize` to find the min. volume ellipsoid) |  7.6 ms         | ±0.1 ms |

---

### Decision · Program_Chairs · 2023-09-21

**Decision:**

Accept (poster)

**Comment:**

I thank the reviewers for their comments. I took the discussion as well as the author feedback into consideration. there seems to be general consensus among the reviewers that the paper is good with positive reviews. I therefore will recommend acceptance of the paper.